# CPEB1 directs muscle stem cell activation by reprogramming the translational landscape

Wenshu Zeng[1,2,3,4,5], Lu Yue [1,2,3,4,5], Kim S. W. Lam [1,2,3,4,5], Wenxin Zhang[1,2,3,4,5], Wai-Kin So[1,2,3,4,5], Erin H. Y. Tse [1,2,3,4,5,6] & Tom H. Cheung [1,2,3,4,5,6,7 ✉]

Skeletal muscle stem cells, also called Satellite Cells (SCs), are actively maintained in quiescence but can activate quickly upon extrinsic stimuli. However, the mechanisms of how quiescent SCs (QSCs) activate swiftly remain elusive. Here, using a whole mouse perfusion fixation approach to obtain bona fide QSCs, we identify massive proteomic changes during the quiescence-to-activation transition in pathways such as chromatin maintenance, metabolism, transcription, and translation. Discordant correlation of transcriptomic and proteomic changes reveals potential translational regulation upon SC activation. Importantly, we show Cytoplasmic Polyadenylation Element Binding protein 1 (CPEB1), post-transcriptionally affects protein translation during SC activation by binding to the 3′ UTRs of different transcripts. We demonstrate phosphorylation-dependent CPEB1 promoted Myod1 protein synthesis by binding to the cytoplasmic polyadenylation elements (CPEs) within its 3′ UTRs to regulate SC activation and muscle regeneration. Our study characterizes CPEB1 as a key regulator to reprogram the translational landscape directing SC activation and subsequent proliferation.

[1] Division of Life Science, The Hong Kong University of Science and Technology, Hong Kong, SAR, China. [2] Center for Stem Cell Research, The Hong Kong University of Science and Technology, Hong Kong, SAR, China. [3] HKUST-Nan Fung Life Science Joint Laboratory, The Hong Kong University of Science and Technology, Hong Kong, SAR, China. [4] State Key Laboratory of Molecular Neuroscience, The Hong Kong University of Science and Technology, Hong Kong, SAR, China. [5] Molecular Neuroscience Center, The Hong Kong University of Science and Technology, Hong Kong, SAR, China. [6] Hong Kong Center for Neurodegenerative Diseases, Hong Kong, SAR, China. [7] Guangdong Provincial Key Laboratory of Brain Science, Disease and Drug Development, Shenzhen-Hong Kong Institute of Brain Science, HKUST Shenzhen Research Institute, Shenzhen, China. ✉email: tcheung@ust.hk

Tissue stem cells are required for tissue homeostasis and repair[1,2]. In low turnover tissues, stem cells are maintained in quiescence for prolonged periods[3,4]. Quiescence is a reversible G0 stage with basal cellular metabolism[5]. Muscle stem cells, or satellite cells (SCs), are indispensable for muscle regeneration[2,6–8]. SCs are tightly maintained in quiescence in uninjured skeletal muscles[9]. Upon stimulation (i.e., acute injury), quiescent SCs (QSCs) can activate rapidly and re-enter the cell cycle, subsequently differentiating and fusing together to repair muscle[10]. Recent studies showed that quiescent signatures could change within hours during isolation[11–13], with freshly isolated SCs (fiSCs) having already acquired an early activation gene signature[14]. Thus, fiSCs previously thought of as quiescent, are indeed at an early phase of activation. While different groups have recently developed methods to capture bona fide quiescent SCs[11–13], the molecular mechanisms regulating the quiescence-to-activation transition are still largely unknown.

Recent studies on early activation are limited to the transcriptome and epigenome[11–13]. However, proteins are the main effectors of functional cellular regulation. In addition, RNA levels may not accurately reflect the abundance of proteins due to post-transcriptional regulations[15,16]. Thus, the use of transcriptomic analysis may not provide adequate information regarding the active signaling effectors that are expressed in QSCs, or during the cell state transition.

Post-transcriptional regulation on a single gene level to manipulate protein synthesis was demonstrated to regulate the SC quiescence-to-activation transition[9]. Some activation-related transcripts are expressed in QSCs but are translationally repressed by post-transcriptional regulation to maintain SC quiescence. For instance, oncogene *Dek* mRNA is highly expressed in QSCs while translation is inhibited by miR-489, a QSC-specific miRNA[17]. *Myf5* transcripts were reported to be sequestered in ribonucleoprotein (mRNP) granules together with miR-31 in QSCs[18]. *Myod1* mRNA is expressed in QSCs while its translation is suppressed by RNA-binding protein Staufen-1[19]. Upon injury, these inhibitions are relieved for rapid protein synthesis to drive SC activation[17–19]. However, how post-transcriptional regulation manipulates the global proteomics landscape to drive the SC quiescence-to-activation transition remains to be explored.

The 3′ UTR of mRNA functions as a post-transcriptional regulation hotspot by harboring a series of motifs such as microRNA (miRNA) target sites, AU-rich elements (AREs), and polyadenylation signals (PASs)[20]. After binding to the target transcript, miRNAs drive the formation of an RNA-induced silencing complex (RISC) by recruiting the Argonaute (Ago) protein to directly cleave the target mRNA or recruit additional proteins to achieve translational repression[21]. Different from miRNA target sites, AREs either induce or suppress protein translation depending on the function of the RNA-binding protein[22]. For instance, the Hu RNA-binding protein family stabilizes their target transcripts resulting in an elevated translational output, whereas AUF1, TTP, BRF1, TIA-1, and KSRP destabilize mRNA and reduce protein expression[22]. Alternative usage of PASs regulates the length of 3′ UTRs, resulting in a differential number of RNA-regulatory motifs, and therefore, varying levels of protein production[23].

Cytoplasmic polyadenylation elements (CPEs)[24], also located on 3′ UTRs, are found in around 20% of mammalian transcripts[25,26]. CPE-binding protein 1 (CPEB1) is an RNA-binding protein that binds to CPE sequences and regulates translation of its target transcripts by inducing cytoplasmic manipulation of their poly(A)-tails[27–30]. After binding to the CPEs, CPEB1 recruits cytoplasmic poly (A) polymerase GLD2 to elongate the poly (A) tail to maintain mRNA stability[31,32]. The stability of mRNAs is positively correlated with translational output[33,34]. CPEB1 regulates cellular function by post-transcriptionally controlling the translation of its targeted transcripts[35]. CPEB1 was reported to promote *Xenopus* oocyte maturation by activating the maternal mRNA translation, including *Cyclin B5* and *Emi1*[29,31]. CPEB1 also regulates human fibroblast senescence and bioenergetics by controlling *TP53* translation[27]. CPEB1 was reported to restrain the proliferation of glioblastoma cells through the regulation of *p27Kip1* mRNA translation and modulates glioma stem cell differentiation via regulating *Hes1* and *Sirt1* translation[36,37]. Besides, CPEB1 controls HeLa cell proliferation and G1 phase entry by regulating the expression of a series of cell-cycle-related genes[38,39]. Cell cycle re-entry is a hallmark of the SC quiescence-to-activation transition[40,41]. However, the genome-wide mRNA targets or the proteome affected by CPEB1 and how CPEB1 is involved in regulating the SC quiescence-to-activation transition are largely unknown.

In this study, we uncover the in vivo QSC proteomics signature and observe a change in the translational landscape during the SC quiescence-to-activation transition. Discordant correlation of the SC transcriptome and proteome suggests the transition from quiescence to activation is regulated post-transcriptionally. We further demonstrate that the translational regulator CPEB1 regulates SC activation and proliferation by reprogramming the translational landscape. In SCs, CPEB1 promotes protein expression via CPEs within the 3′ UTRs in a phosphorylation-dependent manner. Interestingly, the manipulation of CPEB1 phosphorylation affects SC activation, muscle regeneration, and Pax7+ self-renewed SC number. Together, we reveal CPEB1 as a key regulator to establish a translational landscape for SC activation in a post-transcriptional manner.

## Results

**Differential proteomic landscape during SC quiescence exit.** As previously reported, SCs acquire an early activation signature during the isolation process[11,12]. We thus leveraged the in situ fixation technique[13,42] to preserve the bona fide quiescent SCs for analysis (named QSCs in this study). To understand the QSC proteome and its changes during activation, we performed mass spectrometry on QSCs, fiSCs, activated SCs sorted from 3 days injured muscles (iASCs), and cultured activated SCs (cASCs, fiSCs plated down for 48 hours, in the proliferating state) (Fig. 1a). From a whole proteome aspect, QSCs differed from fiSCs while QSCs and fiSCs were more distant from iASCs and cASCs, indicating a rapid and significant activation response of QSCs and, that the fixation approach can preserve the unique proteomic signature of QSCs in vivo (Fig. 1b, protein expression listed in Supplementary Data 1). The proteomic signatures of iASCs and cASCs were similar (Fig. 1b, Supplementary Data 1). To validate the reliability of the proteome, we compared the expression level of various known proteins. For instance, cell cycle proteins (Cdk and Mcm protein families) and activation markers (Myod1, Yap1, Mtor, and Ki67) were not expressed in QSCs and fiSCs, but strongly expressed in cASCs and iASCs, while markers for QSCs, such as Pax7, Calcr, and Numb, were highly expressed in QSCs and fiSCs (Fig. 1c). We also observed a metabolism reprogramming from fatty acid oxidation to glycolysis during SC activation (Fig. 1d), in agreement with reported transcriptome analysis[43]. Intriguingly, we observed differential expression of transmembrane protein/receptors during SC activation (Fig. 1e), indicating that SCs reprogram interactions with the niche for proper activation. Together, our data presented a whole proteome view of QSCs and revealed proteins that are differentially expressed during SC activation.

To decipher the dynamic proteome changes during SC activation, we compared the proteomes of QSCs, fiSCs, and cASCs in a pairwise manner and performed functional

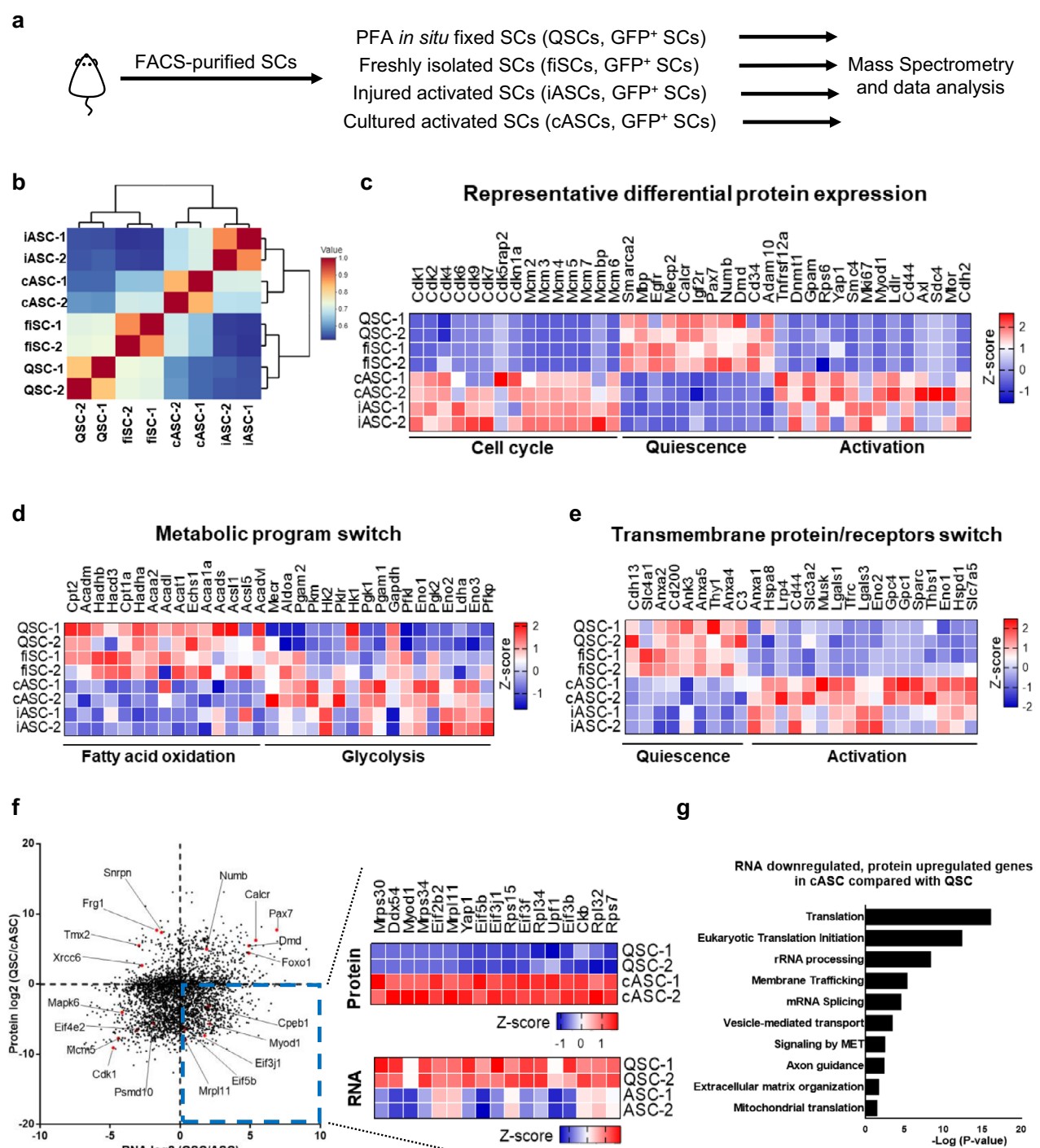

**Fig. 1 Discordant proteomic and transcriptomic signatures reveal a potential translational control mechanism during the SC quiescence-to-activation transition. a** Schematic illustration of the workflow for the proteomic analysis of FACS-isolated SCs. The SCs were sorted from Pax7-nGFP mice. QSCs were sorted from 0.5% PFA-perfused mice. fiSCs were sorted from unperfused mice. Injured ASCs (iASCs) were sorted from 3 days 1.2% BaCl₂ injured mice. Cultured ASCs (cASCs) were 2 days cultured SCs. (*n* = 2 independent experiments). **b** Hierarchical clustering of the protein expression (in the number of spectra) for QSCs, fiSCs, iASCs, and cASCs. **c** Heatmap of representative expression for cell cycle, quiescence, and activation-specific proteins of QSCs, fiSCs, iASCs, and cASCs. **d** Heatmap of representative expression for fatty acid oxidation and glycolysis-related proteins of QSCs, fiSCs, iASCs, and cASCs. **e** Heatmap of representative expression for transmembrane protein/receptors of QSCs, fiSCs, iASCs, and cASCs. **f, g** Analysis of transcriptome and proteome on QSCs and cASCs (proteome comparison) and iASCs (transcriptome comparison). **f** Scatter plot of the fold change of RNA and protein comparing QSCs to ASCs. Heatmap shows the RNA and protein expression of representative genes in the lower right quadrant of the scatter plot. **g** Functional enrichment analysis of genes from the lower right quadrant of **f** by g:Profiler[99]. The g:Profiler uses a hypergeometric test to measure the significance of functional terms in the input gene list.

enrichment analysis on the differentially expressed proteins. In the cell cycle and DNA replication cluster, we observed that "packaging of telomere ends" and "nucleosome assembly" pathways were highly enriched in QSCs when compared with fiSCs or cASCs (Supplementary Fig. 1a, c, e). The enriched pathways in fiSCs, compared to QSCs, were "Orc1 removal from chromatin" and "DNA replication pre-initiation" (Supplementary Fig. 1b), revealing that SCs initiates the cell cycle program rapidly after its dissociation from the niche. Intriguingly, proteins related to cristae formation, such as Atp5mg, Atp5f1b, and Atp5f1d, were strongly expressed in QSCs specifically (Supplementary Fig. 1a), suggesting that SCs coordinate subcellular organelles for metabolism reprogramming from oxygen consumption reactions (fatty acid oxidation and TCA cycle) to non-oxygen required reactions (glycolysis) upon SC activation since the cristae in the mitochondrial inner membrane provides a larger chemical reaction area, enabling efficient oxidative phosphorylation[44,45]. Oxygen consumption reactions produce reactive oxygen species (ROS) such as superoxide or hydrogen peroxide, which causes DNA damage[46,47]. DNA repair categories such as "base excision repair", "DNA double-strand break response", and other repair pathways, were highly enriched in QSCs (Supplementary Fig. 1a), while pathways for repairing DNA damage during DNA replication such as "p53-dependent G1/S DNA damage checkpoint" and "gap-filling DNA repair", were enriched only in cASCs (Supplementary Fig. 1d, f). These suggest that DNA repair mechanisms are altered due to DNA damage in QSCs and ASCs. Together, we present a blueprint of SC proteomes during SC activation, and the observed changes of protein expressions suggest that a significant change in the translational landscape is required for SC activation and proliferation.

**Discordant proteomic and transcriptomic signatures in SCs.** We and others have previously reported the transcriptomic changes during the SC quiescence-to-activation transition[11–13]. To correlate between the proteome and the transcriptome signatures during the SC quiescence-to-activation transition, we visualized the differences of these signatures on a scatter plot (Fig. 1f, the fold changes of transcripts and proteins are listed in Supplementary Data 2). On the lower left and the upper right quadrants of this scatter plot, we observed concordant upregulated or downregulated transcripts and proteins during SC activation. For example, genes that are essential for the maintenance of stemness or quiescence such as *Pax7*, *Calcr*, and *Numb*, are downregulated during the transition. On the other hand, genes involved in translation or the cell cycle such as *Eif4e2*, *Cdk1*, and *Mcm5*, are upregulated during the transition. Intriguingly, we observed a large number of genes in the lower right quadrant, which represented genes with highly expressed transcripts and low protein levels in QSCs. During SC activation, the proteins of these genes are highly expressed, despite their transcripts being downregulated (Fig. 1f). Examples of these genes are involved in regulating myogenic lineage progression or translational regulation such as *Myod1*, *Mrpl11*, *Rps15*, and *Eif5b*. To further investigate the gene functions in this quadrant, we performed functional enrichment analysis and found that pathways such as "Translation", "Eukaryotic Translation Initiation", and "rRNA Processing", are significantly enriched (Fig. 1g). Thus, the discordant proteomic and transcriptomic signatures suggest a potential translational control mechanism during the SC quiescence-to-activation transition.

**CPEB1 is required for SC activation and proliferation.** As previously shown, discordant transcriptomic and proteomic signatures suggest that the quiescence-to-activation transition is post-transcriptionally regulated. Previous studies showed that CPEB1 is a master regulator of translational control by acting on CPEs on 3′ UTRs during *Xenopus* oocyte maturation[31,32]. We next tested the hypothesis of whether CPEB1 functions to regulate this transition (Fig. 2a). CPEB1 expression was abundant and detectable in both QSCs and early activated SCs (cultured for 2 hours), and significantly upregulated in ASCs (Fig. 2b, c and Supplementary Fig. 3a–c, CPEB1 antibody specificity is shown in Supplementary Fig. 2a, b, d). The majority of CPEB1 is localized in the cytoplasm of QSCs and ASCs. Interestingly, QSCs on paraformaldehyde (PFA)-perfused, fixed fibers have more extended cytoplasmic projections compared with early activated SCs (Fig. 2b). These observations were consistent with the in vivo imaging of SCs after tissue clearing[48], supporting the notion that our fixation approach captured the in vivo signature of SCs. To test whether CPEB1 regulates SC activation, we knocked down CPEB1 expression during SC activation (the efficiency and specificity of siRNA for CPEB1 is shown in Supplementary Fig. 2c-h). EdU was added to mark cycling SCs during the course of in vitro SC culture (Fig. 2d). Knocking down CPEB1 delayed SC activation in both fiber-associated SCs (Fig. 2e, f) and Fluorescent Activated Cell Sorting (FACS)-purified SCs (Fig. 2g, h), suggesting that CPEB1 is essential for SC activation. The overall SC number (Syn4 marks both QSCs and ASCs) was also decreased after CPEB1 knockdown (Supplementary Fig. 3d–h). To test whether the reduced SC number is caused by a defect in SC proliferation, we cultured fiSCs for 30 hours and then pulsed these activated SCs with EdU for 6 hours before harvest (Supplementary Fig. 3i). We also evaluated the expression of the cell cycle marker Ki67. Both experiments showed that the proliferation of SCs was inhibited when CPEB1 was knocked down (Supplementary Fig. 3j-o). Together, these suggested that CPEB1 is essential for both SC activation and proliferation.

To understand whether CPEB1 knockdown affects SC cell fate decisions, we analyzed the expression of Pax7, Myod1 and MyoG in SCs. Compared to the control, the percentages of Pax7, Myod1, and MyoG expressing SCs did not change significantly (Supplementary Fig. 4a-d, f, g), indicating the loss of CPEB1 did not alter SC cell fate determination. Interestingly, we observed decreased Myod1 protein levels after CPEB1 knockdown (Supplementary Fig. 4e), further supporting the notion that CPEB1 regulates SC activation since Myod1 protein is correlated to SC activation[19,49]. To determine whether CPEB1 siRNA-treated SCs can eventually re-enter the cell cycle, we cultured SCs with EdU for 72 hours after CPEB1 knockdown, a time point by which the effect of siRNA is diminishing. EdU was incorporated into the CPEB1 siRNA-treated SCs (Supplementary Fig. 5c, d). To assess whether CPEB1 knockdown induces apoptosis, we stained SCs for cleaved Caspase 3 after CPEB1 knockdown and found that the loss of CPEB1 in SCs did not induce apoptosis (Supplementary Fig. 5b, the cleaved Caspase 3 antibody specificity is shown in Supplementary Fig. 5a).

**CPEB1 binds to transcripts to control translation.** To investigate the targets of CPEB1 and their corresponding functions during SC early activation, we performed a CPEB1 RNA Immunoprecipitation Sequencing (RIP-seq) experiment using fiSCs (Fig. 3a, CPEB1 IP control western blot is shown in Supplementary Fig. 6a). To identify CPEB1-associated transcripts, we compared the CPEB1 IP group to the IgG control group and identified 1561 transcripts associated with CPEB1 that are potential CPEB1 targets (Fig. 3b, c, Supplementary Data 3, Fold change > 2, DEseq2, P-adj < 0.05). To further confirm the specificity of CPEB1 RIP-seq, we analyzed the percentage of CPEs-containing genes in CPEB1-associated and IgG-associated genes

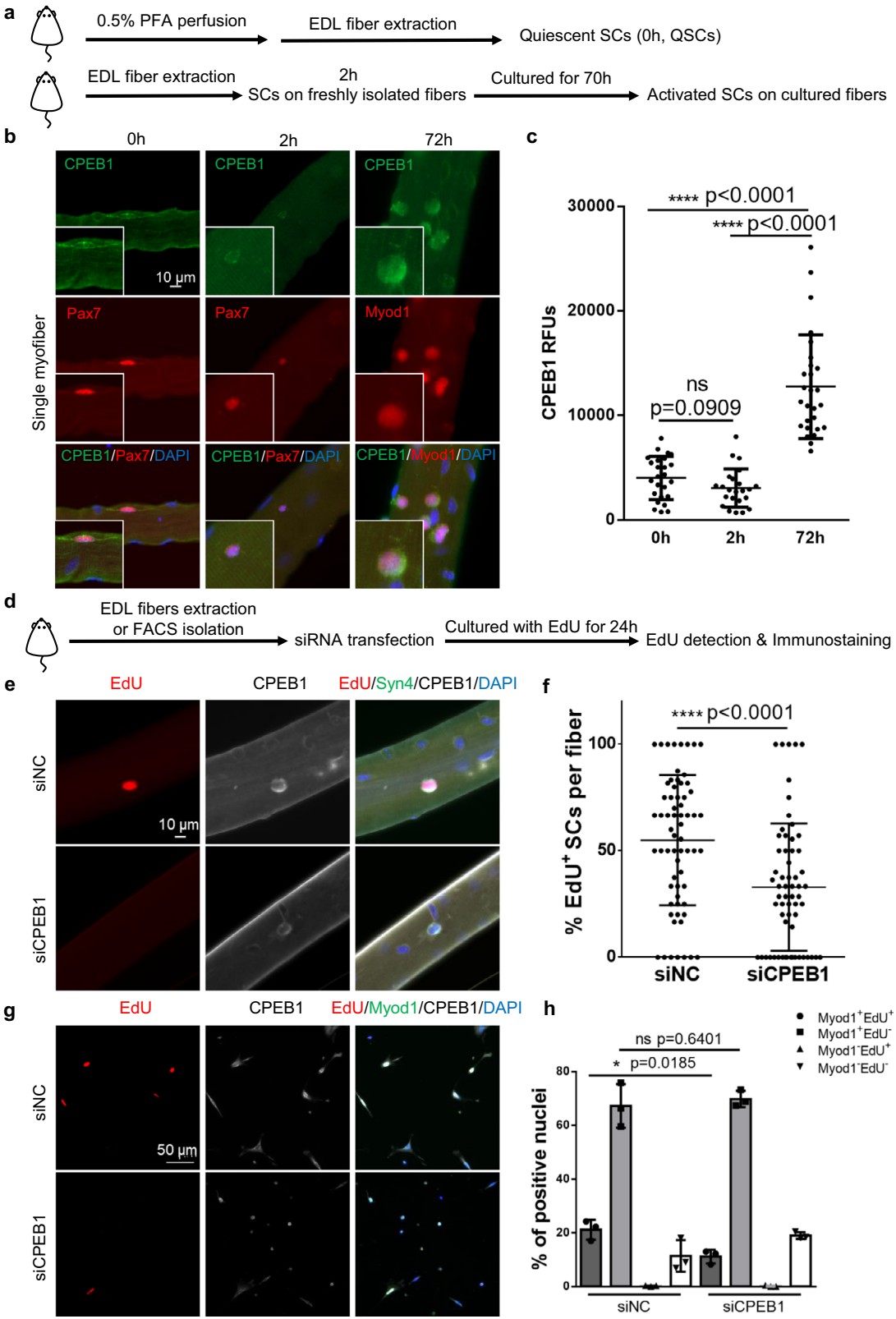

and found that over 60% of CPEB1-associated genes contain CPEs while less than 10% of IgG bound transcripts contain CPEs (Fig. 3d). In summary, this CPEB1 RIP-seq identified mRNAs that are directly bound by CPEB1 during SC early activation.

To study the function of CPEB1-associated genes, we performed functional enrichment analysis. These transcripts are enriched in energy metabolism, spliceosome, ribosome

biogenesis, RNA transport, and TCA-cycle-related pathways (Fig. 3e). Regarding spliceosome-associated pathways, transcripts of genes involved in pre-mRNA splicing such as *Dhx, Rbm,* and *Snrp* gene families are bound by CPEB1 (Fig. 3f). In addition, for the energy metabolic pathways, genes encoding proteins involved in oxidative phosphorylation or TCA cycle identified as CPEB1 targets include *Cox5a, Ndufa11, Uqcr10, Mdh1, and Sdha*

**Fig. 2 CPEB1 is highly expressed in ASCs and required for SC activation. a–c** CPEB1 protein expression analysis on fiber-associated QSCs and ASCs. **a** Schematic illustration of the experimental design to obtain fiber-associated QSCs and ASCs. **b** Immunostaining of CPEB1 on QSCs (marked by Pax7 protein) and ASCs (marked by Myod1 protein). Nuclei were stained with DAPI. **c** Quantification of CPEB1 relative fluorescence units (RFUs) of SCs per myofiber. ($n = 3$ independent experiments, the number of quantified fibers is 25, 23, 27 for 0 h, 2 h, 72 h fiber staining experiments respectively). **d–h** EdU incorporation analysis on SCs after CPEB1 knockdown. **d** Schematic illustration of the EdU incorporation analysis on CPEB1 knocked down fiber-associated SCs. **e** 24 hours after siRNA transfection, fiber-associated SCs were harvested for EdU detection and stained for CPEB1 and Syndecan 4 (Syn4). SCs are marked by Syn4. Nuclei were stained with DAPI. **f** Quantification of EdU+ SCs per fiber. ($n = 4$ independent experiments, the number of quantified fibers is 65 and 62 for siNC and siCPEB1 groups, respectively. **g** 24 hours after siRNA transfection, SCs were harvested for EdU detection and stained for Myod1 and CPEB1. Nuclei were stained with DAPI. **h** Quantification of MyoD+EdU+, MyoD+ EdU−, MyoD− EdU+, and MyoD−EdU− SCs after siRNA transfection. ($n = 3$ independent experiments). Data are presented as mean ± SD in panels **c, f, h**. The $p$-values calculated by two-tailed unpaired $t$-test are used for comparing two groups in panels **c, f, h**. Source data are provided as a Source Data file.

(Fig. 3g). During SC activation, there is an indispensable need to establish the essential transcriptional and translational programs to boost energy production and macromolecules for cell growth and cell cycle re-entry[50–52]. Taken together, our results suggest that CPEB1 is a post-transcriptional regulator for the reprogramming of the SC cellular environment during SC activation.

In addition, CPEB1 also binds to transcripts that encode proteins involved in translational regulation such as genes for ribosome biogenesis and RNA transport. Regarding ribosome biogenesis, transcripts of genes like *Utp15, Wdr3, Wdr43, Imp3,* and *Imp4*, are also among the transcripts bound by CPEB1 (Fig. 3h). Interestingly, CPEB1 is also associated with the *Eif* family of transcripts for example, *Eif1a, Eif2s1, Eif3a, Eif4b,* and *Eif5b*, which are essential for mRNA targeting to the ribosome[53] (Fig. 3h). Taken together, these observations indicated that CPEB1 is a key regulator to control the translational landscape by targeting protein synthesis machinery post-transcriptionally.

**CPEB1 reprograms the translational landscape in SCs**. To explore the effect of CPEB1 on protein expression, we performed mass spectrometry on SCs receiving control (siNC) or CPEB1 siRNA treatments (siCPEB1) (Fig. 4a). CPEB1 knocked down SCs have a distinct proteome compared with the controls, suggesting CPEB1 can modulate the SC translational landscape (Fig. 4b). The expression of cell cycle-related proteins, such as the Cdk and Mcm protein families, were downregulated after CPEB1 siRNA treatment (Fig. 4c). This is consistent with the defect we observed during SC activation and proliferation when CPEB1 is knocked down (Fig. 2 and Supplementary Fig. 3). We identified 518 downregulated proteins after CPEB1 knockdown (Fig. 4d and Supplementary Data 4). To uncover the functions of these CPEB1-dependent proteins, we performed functional enrichment analysis and found that these proteins are enriched in functional categories such as "mRNA stability", "MAPK6/MAPK4 signaling", "cellular response to hypoxia", "cell cycle", "myogenesis", and "translation" (Fig. 4e–g). Intriguingly, CPEB1-dependent proteins were also enriched in the "processing of capped intron-containing pre-mRNA" (Fig. 4e and Supplementary Fig. 7a). To identify the CPEB1-dependent proteins during SC activation, we compared the lists of CPEB1-associated genes, the upregulated proteins in cASCs, and the CPEB1-dependent proteins. We identified 67 CPEB1-dependent proteins that were upregulated during SC activation (Fig. 4h), of which 52 of the 67 genes contain CPEs in their 3′ UTRs (Fig. 4i), confirming the specificity of CPEB1 RIP-seq. Interestingly, the functional enrichment analysis identified 12 of 67 genes were enriched in translation regulation pathways (Fig. 4j, k).

Previously, we showed a discordant protein and transcript expression profile during SC activation, suggesting a mode of post-transcriptional regulation (Fig. 1f). To further investigate whether CPEB1 mediates the post-transcriptional regulation, we mapped the CPEB1-associated transcripts on the scatter plot that showed the changes of transcriptomic and proteomic data during SC activation (Supplementary Fig. 7b). We observed that some of the CPEB1-associated transcripts are downregulated during the SC quiescence-to-activation transition, but its protein expressions are upregulated. These genes are essential for myogenic lineage progressions such as the myogenic factor *Myod1* and translation machinery such as the genes encoding the translation initiation complex (e.g., *Eif5b* and *Eif3g*). Interestingly, selective translation factors, such as *Eif3a* and *Eif4a*, appear to require sustained protein expression with its transcripts also being highly expressed in ASCs. Together, our data indicate that CPEB1 functions to reprogram the translation landscape for SC activation in a post-transcriptional manner.

**CPEB1 regulates Myod1 protein output by acting on CPEs**. To investigate how CPEB1 regulates translation post-transcriptionally, we focused on *Myod1*, one of the 67 genes that was both a CPEB1-dependent and upregulated protein in ASCs (Fig. 4h). Myod1 was the only myogenic factor with transcripts enriched significantly by CPEB1 immunoprecipitation (Fig. 5a, c). We also reconfirmed the binding of CPEB1 proteins to *Myod1* transcripts by CPEB1 RNA-IP-qPCR (Fig. 5b). Intriguingly, when detecting CPEB1 expression patterns in SCs, we observed that CPEB1 existed as cytoplasmic puncta in SCs during early activation (Fig. 5d). *Myod1* transcripts were transported from the nuclei to the cytoplasm during the SC quiescence-to-activation transition, and the timing of translation coincided with Myod1 protein expression (Supplementary Fig. 8a–e). Based on these observations, we hypothesize that CPEB1 regulates the translation of *Myod1* transcripts to drive the SC quiescence-to-activation transition. CPEB1 is reported to bind to and regulate translation of CPE-containing transcripts[29,31,54]. An examination of the *Myod1* 3′ UTR revealed two canonical CPEs. To understand whether CPEB1 binds to the two CPEs on the *Myod1* 3′ UTR, we performed a luciferase assay on 293 T cells co-transfected with a CPEB1 expressing construct and a luciferase vector containing the wild-type or mutated-CPE *Myod1* 3′ UTR (Fig. 5e). Ectopic CPEB1 expression increased the luciferase activity of the wild-type 3′ UTR construct, whereas this effect was abolished when the CPEs were mutated (Fig. 5e). These results suggest that CPEB1 post-transcriptionally regulates Myod1 translation by acting on the CPEs within its 3′ UTR.

To examine whether CPEB1 influences Myod1 protein level in SCs in vivo, traditional siRNA approaches may not be fast enough as SCs activate and express Myod1 protein rapidly upon dissociation from its niche (Supplementary Fig. 8d–h). Instead, we opted to block CPEB1 function by transfecting a CPEB1 antibody into freshly isolated SCs (Fig. 5g). We reasoned that the use of a neutralizing antibody would provide an instant effect, whereas the use of siRNA will require minimally 16–24 hours to function[55]. To confirm the transfected CPEB1 antibody indeed neutralized CPEB1 protein function, we performed a luciferase assay on 293T cells co-transfected with a CPEB1 expressing

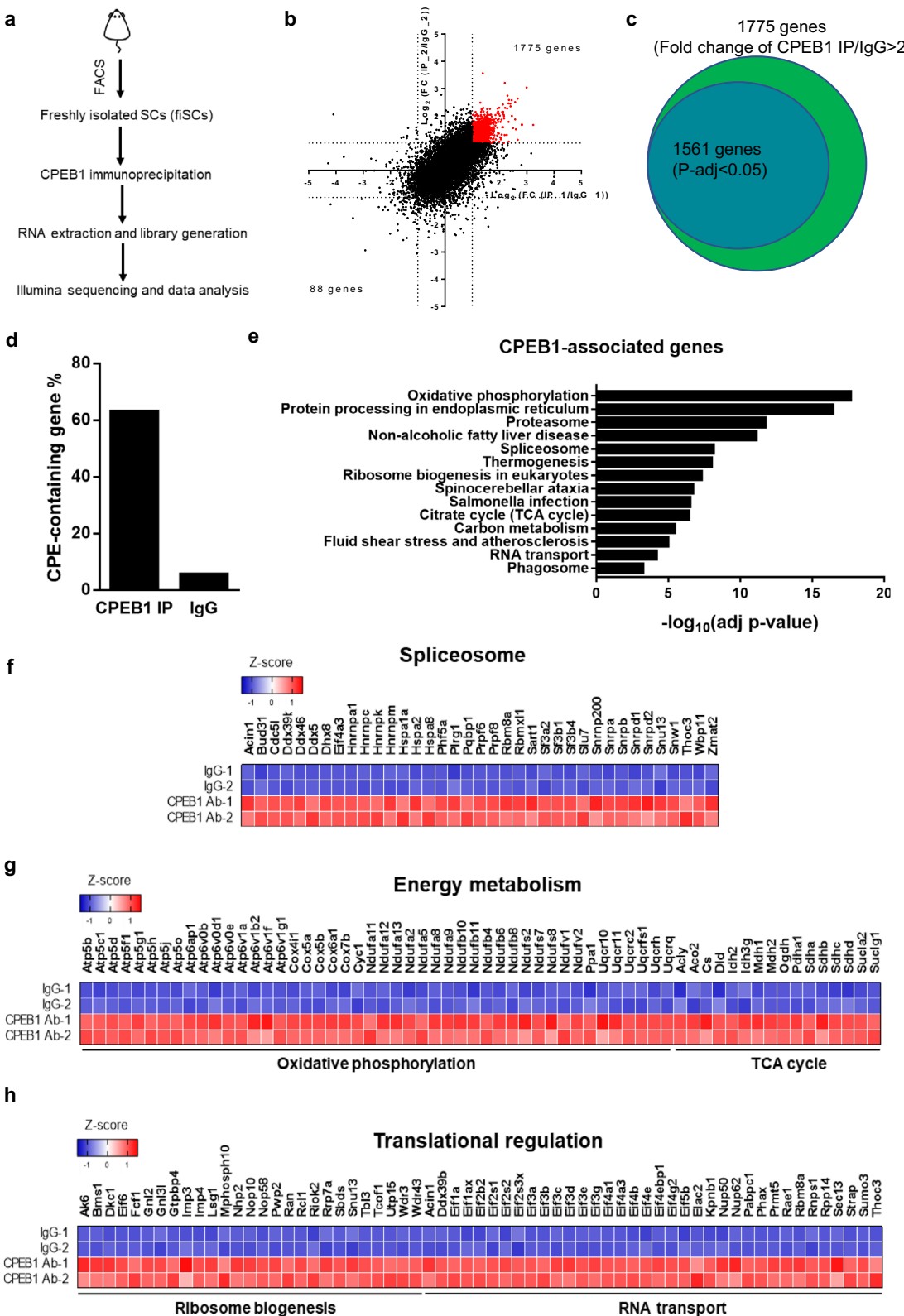

construct and a luciferase vector containing the Myod1 3′ UTR, followed by CPEB1 antibody transfection. The data showed that the CPEB1 antibody was able to neutralize CPEB1 function in promoting Myod1 protein expression in vitro (Fig. 5f). Compared with the control, SCs transfected with CPEB1 antibody resulted in a significantly lower number of Myod1-expressing SCs with reduced Myod1 expression levels (Fig. 5h–j). This suggests that

CPEB1 is required for the rapid expression of Myod1 protein during the SC quiescence-to-activation transition.

**CPEB1 phosphorylation is required to regulate translation.** CPEB1 phosphorylation is required for its function to regulate translational control of different biological processes[27,30,56–58]. To

**Fig. 3 CPEB1 regulates transcripts encoding for proteins involved in transcriptional, translational, and energy metabolism during SC quiescence-to-activation transition.** **a–d** CPEB1-associated genes analysis using RNA immunoprecipitation sequencing (RIP-seq). **a** Schematic illustration of the workflow of the CPEB1 RIP-seq. **b** Scatter plot of the genes detected in two independent CPEB1 RIP experiments. Each dot represents one gene. The X- and Y-axis represent the fold change of each CPEB1 RIP experiment by normalizing to IgG controls. (n = 2 independent experiments). **c** Venn diagram showing the genes significantly enriched after CPEB1 RIP when compared with IgG controls (Fold change (CPEB1-Ab-IP/IgG) > 2, DEseq2, P-adj < 0.05). DESeq2[94] is a method to perform differential analysis of count data, using shrinkage estimation for dispersions and fold changes. **d** Bar graph showing percentage of CPE-containing genes in CPEB1-associated genes and IgG-associated genes. **e** Functional analysis of CPEB1-associated genes in KEGG pathways using g:Profiler[99]. The g:Profiler uses a hypergeometric test to measure the significance of functional terms in the input gene list. **f–h** Heatmaps of the percentile rank of representative CPEB1-associated genes involved in **f** spliceosome, **g** energy metabolism, and **h** translational regulation.

explore the role of CPEB1 phosphorylation in SCs, we examined whether CPEB1 is phosphorylated in SCs by utilizing a phosphorylated CPEB1 antibody (pCPEB1). The pCPEB1 antibody specificity was demonstrated by immunostaining and western blot of lysates from wild-type or mutated CPEB1 overexpressing cells, western blot of lysates from Lambda (λ) phosphatase-treated cells and Enzyme-Linked Immunosorbent Assay (ELISA) for phosphorylated peptides or non-phosphorylated peptides (Supplementary Fig. 9a–d). Notably, pCPEB1 was undetectable in QSCs but became detectable in early activated SCs (cultured for 2 hours) and was upregulated in ASCs (Supplementary Fig. 10a–e). This indicates that CPEB1 becomes phosphorylated during the SC quiescence-to-activation transition.

Previously, we showed that the majority of CPEB1 existed as cytoplasmic puncta signals in QSCs and fiSCs (Fig. 5d). We also observed that pCPEB1 existed as cytoplasmic puncta by confocal imaging (Fig. 6a). To validate that pCPEB1 regulates translation through CPEs, we co-transfected 293 T cells with the wild-type, or phosphorylation mutant CPEB1 (T171A, S177A) constructs together with the luciferase vector containing the wild-type or the mutated-CPE *Myod1* 3′ UTR. No beneficial translational effect was detected when the phosphorylation mutant CPEB1 was expressed nor when the CPEs were mutated (Fig. 6b), highlighting the importance of the role of CPEB1 phosphorylation in the post-transcriptional control of Myod1 protein expression. To confirm whether phosphorylation of CPEB1 is required for Myod1 protein expression in SCs, we infected SCs with adenovirus containing either the wild-type or phosphorylation mutant CPEB1 coding sequence. The Myod1 protein level was elevated significantly in SCs overexpressing the wild-type CPEB1. Conversely, it was significantly reduced with the phosphorylation mutant (Fig. 6c, d). Thus, our data suggest that CPEB1 phosphorylation is essential for *Myod1* translation in SCs.

To investigate how the phosphorylation of CPEB1 is involved in translational regulation, we examined whether the phosphorylation mutant affects the capability of CPEB1 to bind mRNAs. We overexpressed CPEB1-mVenus or CPEB1 (T171A, S177A)-mVenus fusion protein in C2 cells and then performed mVenus immunoprecipitation followed by qRT-PCR analysis. The mRNA binding ability was not affected after phosphorylation site mutation (Fig. 6e). We next tested whether the phosphorylation mutant affects the interaction of CPEB1 with the translational machinery. To test this hypothesis, we performed mVenus immunoprecipitation followed by mass spectrometry (Fig. 6f). The CPEB1 peptides were only detected in mVenus immunoprecipitation samples while being undetectable in IgG immunoprecipitation samples, indicating this assay is reliable in detecting CPEB1 interacting proteins (Fig. 6g, h). We obtained 66 CPEB1-bound proteins and 99 CPEB1 (T171A, S177A)-bound proteins (Supplementary Fig. 11a, b and Supplementary Data 5). Among these proteins, 29 proteins were specifically interacting with CPEB1, while 62 proteins were interacting with CPEB1 (T171A, S177A) (Fig. 6i). To study the functional networks of CPEB1 interacting proteins, we performed functional proteins association

network analysis using STRING. The network of these proteins was distinguished (Supplementary Fig. 11c–e). The proteins that specifically interacted with CPEB1 function to regulate proteasomes and ribosomes while the proteins only interacting with CPEB1 (T171A, S177A) are involved in endocytosis and lysosomes (Fig. 6j). Proteins interacting with both CPEB1 and CPEB1 (T171A, S177A) regulate proteasomes and mRNA surveillance (Fig. 6j). These data suggest that phosphorylation is essential for CPEB1 to regulate translation by interacting with ribosomal proteins.

**CPEB1 phosphorylation regulates SC function.** CPEB1 has been reported to be phosphorylated by Aurora kinase A (Aurka)[27,30,56–58]. To explore the expression pattern of Aurka in SCs, we performed Aurka immunostaining and found that Aurka expression was undetectable in QSCs, low in early activated SCs (cultured for 2 hours) but increased significantly in ASCs (Supplementary Fig. 12a–d). To explore whether CPEB1 is phosphorylated by Aurka in SCs, we treated SCs with the Aurka specific inhibitor MK5108 or activator insulin[59]. Whereas MK5108 is a highly specific inhibitor of Aurka, Aurka can be activated by insulin via the PI3-K pathway[59]. SCs treated with either MK5108 or insulin exhibited the corresponding decreased or increased levels of pCPEB1 (Supplementary Fig. 12e–h). To confirm whether CPEB1 phosphorylation is positively correlated with Aurka activity in SCs, we analyzed the immunofluorescence intensity of pCPEB1 in SCs treated with MK5108 and/or insulin (Supplementary Fig. 12i). Therefore, these data suggest that Aurka could be a regulator for CPEB1 phosphorylation in SCs.

To determine whether CPEB1 phosphorylation is essential for SC activation, we analyzed the relationship between SC activation and CPEB1 phosphorylation. Phosphorylated Aurka (pAurka) and pCPEB1 were positively correlated with Myod1 protein expression during SC early activation (Supplementary Fig. 13a–c). We then manipulated CPEB1 phosphorylation pharmacologically. Using the Aurka inhibitor, we showed that the inhibition of CPEB1 phosphorylation resulted in a decreased percentage of Myod1+ SCs during early activation (Supplementary Fig. 13d–g). We further showed that Aurka inhibitor treatment reduced the percentage of EdU+ SCs (Fig. 7a, b). On the contrary, enhancing CPEB1 phosphorylation using insulin promoted SC activation (Fig. 7a, b). Similar results were observed when fiSCs were transduced with a CPEB1 phosphorylation mutant adenovirus (Supplementary Fig. 13h, i). fiSCs treated with both MK5108 and insulin resulted in a loss of insulin effect on SC proliferation (Fig. 7a, b), suggesting the proliferative effect of insulin is in part mediated by Aurka downstream targets such as CPEB1. Together, these data support the notion that CPEB1 phosphorylation is important for SC proper activation and proliferation.

To examine whether insulin or MK5108 regulates SC activation via CPEB1 phosphorylation, we overexpressed CPEB1 (T171D, S177D), a phosphorylation mimic in SCs with MK5108 treatment or knocked down CPEB1 in SCs with insulin treatment. We observed that the CPEB1 phosphorylation mimic partially

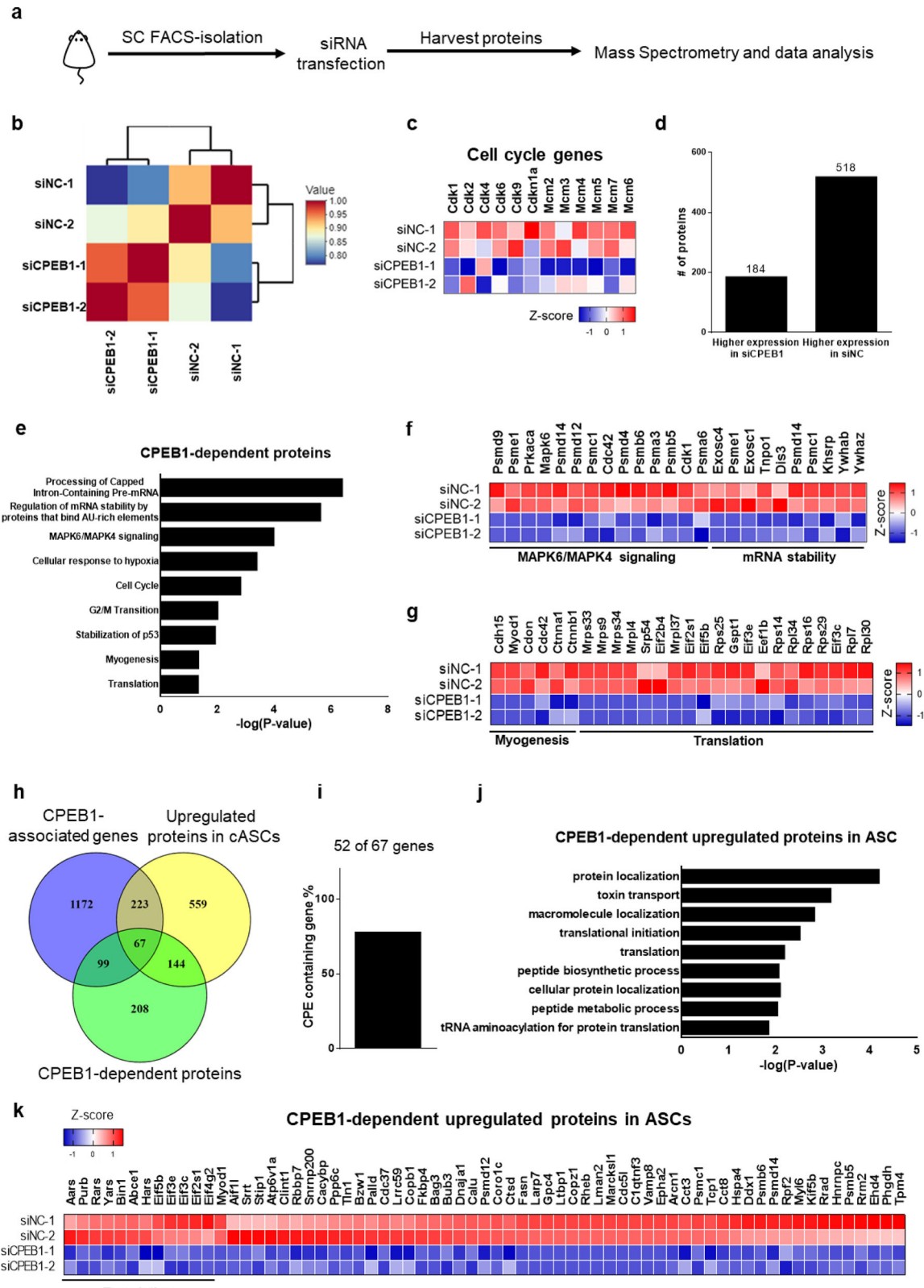

rescued MK5108-mediated proliferation defects (Supplementary Fig. 14a, b). We further showed that CPEB1 knockdown-associated SC activation defects could partially be rescued by insulin treatment (Supplementary Fig. 14c, d). Taken together, we showed that CPEB1 phosphorylation could be in part responsible for the observed effects when SCs were treated with MK5108 or

insulin. However, the results are correlative and the effects are likely mediated by other CPEB1-independent mechanisms.

To further address the role of CPEB1 phosphorylation in vivo in the context of muscle regeneration and Pax7[+] SC number, we modulated CPEB1 phosphorylation in injured Tibialis Anterior (TA) muscle by injecting MK5108 and/or insulin intramuscularly (Fig. 7c). To determine whether MK5108 or insulin injection

**Fig. 4 CPEB1 regulates the translational landscape for SC activation. a–e** CPEB1-dependent proteome analysis. **a** Schematic illustration showing the workflow of proteomic analysis on CPEB1 knocked down SCs. **b** Hierarchical clustering of the protein expression (in the number of spectra) of SCs after CPEB1 knockdown. **c** Heatmap of representative protein expression for cell cycle genes. **d** Quantification of differentially expressed proteins after CPEB1 knockdown. **e** Functional enrichment analysis of CPEB1-dependent proteins using Reactome[100]. ($n = 2$ independent experiments). In Reactome the method used to calculate the statistical significance is the binomial test. **f, g** Heatmaps of representative protein expression involved in MAPK6/MAPK4 signaling, mRNA stability regulation, myogenesis, and translation identified in **e**. **h** Venn diagram of CPEB1-associated genes, upregulated proteins in cASCs, and CPEB1-dependent proteins. **i** Bar graph showing percentage of CPE-containing genes of the 67 proteins in the intersect of **h**. **j** Functional enrichment analysis of the 67 proteins in Biological Process (BP) pathways by g: Profiler[99]. The g:Profiler uses a hypergeometric test to measure the significance of functional term in the input gene list. **k** Heatmap of the protein expression of the 67 proteins in the intersect of **h**.

affects CPEB1 phosphorylation in vivo, we FACS-isolated ASCs from the injured TAs receiving either treatment followed by immunostaining for pCPEB1. The pattern of pCPEB1 immunofluorescence intensity was consistent with MK5108 and/or insulin treated SCs in vitro, suggesting intramuscular injection of MK5108 and/or insulin can affect CPEB1 phosphorylation in vivo (Supplementary Fig. 12j, k). By analyzing the regenerated fiber size, we found that injecting insulin which increases CPEB1 phosphorylation improved muscle regeneration, whereas injection of MK5108, which inhibits CPEB1 phosphorylation dampened muscle regeneration (Fig. 7d–f). We observed an increased number of Pax7$^+$ SCs in regenerated muscle after injecting insulin, whereas injection of MK5108 had the opposite effect (Fig. 7g–i). Taken together, these data indicated that muscular injection of insulin or MK5108 to manipulate CPEB1 phosphorylation is important for SC activation, Pax7$^+$ SC number, and muscle regeneration, providing a potential drug target for improving muscle regeneration and Pax7$^+$ SC number.

## Discussion

In this report, we revealed a significant change in the SC proteomic landscape during the quiescence-to-activation transition. We demonstrated CPEB1 regulates SC activation by directly mediating the translational regulation of its targets in a phosphorylation-dependent manner via CPEs. Furthermore, CPEB1 phosphorylation was critical for muscle regeneration. Together, our data suggest the notion that CPEB1 reprograms the translational landscape during SC activation and muscle regeneration.

SCs are typically isolated for study by FACS, with the sample preparation process includes mechanical dissociation and enzyme digestion. From dissection to cell sorting, the whole process usually takes more than 4 h[60]. Currently, studies on the quiescence-to-activation transition focus on comparing fiSCs and ASCs, which may not give a bona fide quiescent picture as fiSCs were shown to have signs of early activation, such as expression of Myod1 protein and downregulated Pax7 mRNA level[11,13]. Thus, the proteomics landscape of freshly isolated SCs[61] cannot precisely capture the in vivo quiescence signature and the rapid proteomic response during the quiescence-to-activation transition. We and others have developed methods to capture the signature of QSCs[11–13]. In eukaryotes, proteins are the key executors to guide cellular activities. However, because of post-transcriptional regulations, the transcriptomic signature cannot precisely represent the proteomic signature. Therefore, we modified our in situ fixation technique to recover the fixed peptides and subjected the peptides to trapped ion-mobility spectrometry Time-of-Flight (timsTOF) Mass Spectrometry (MS), which is known for its high sensitivity and specificity. Due to the presence of formaldehyde-induced protein cross-links and modifications[62,63], these fixed peptides are difficult to dissociate and characterize by MS, so we reversed the crosslinking by heating the protein lysate at 70 °C for 2 hours with 2% SDS. To our knowledge, this is the initial report to reveal the proteome of

in situ fixed QSCs. Using this approach, we found that the QSC proteome underwent a massive change during SC activation, including changes in DNA methylation, metabolism, cell cycle, and other signaling transduction pathways.

Notably, we observed that Myod1 protein was detected in 20–30% of fiSCs by immunostaining while it was undetectable by MS in our fiSC sample. Of note, we identified a total of 2500–3500 proteins in QSCs. Unlike RNA-seq which could be subjected to library amplification, peptides retrieved from the samples cannot be amplified, limiting our ability to detect low abundant proteins. For instance, CPEB1 protein was only detectable in ASCs by MS, but detected in both QSCs and fiSCs by immunostaining. Thus, our proteomic analysis of the SC quiescence-to-activation transition can only serve as an initial blueprint of the SC proteomes, and an improved proteomics approach is required to fully elucidate the complete proteomic signatures, including the low abundant proteins.

CPEB1 is reported to regulate oocyte maturation, cancer progression, cell cycle, and neural development by controlling translation of various transcripts[31,39,64,65]. CPEB1 was previously shown to modulate proliferation. Its function is context-specific, depending on the target genes and cell types. Here, we showed that CPEB1 was required for SC activation and proliferation. In CPEB1 knockout (KO) mice, germ cell progression stopped before the first meiotic prophase as mRNAs encoding SCP1 and SCP3 were neither polyadenylated nor translated[66]. In synchronized HeLa cells, Cdc20 has a higher protein expression level due to a higher translation efficiency provided by a longer poly (A) tail induced by CPEB1[39]. Interestingly, CPEB1 KO embryonic fibroblasts were shown to bypass senescence and are immortal due to CPEB1-dependent de-repression of Myc[67].

Subcellular localization is also important for the regulatory function of CPEB1. For instance, CPEB1 confers the function of poly(A) tail elongation by recruiting GLD2 in the cytoplasm[68]. A longer poly(A) tail increases mRNA stability and thus enhances protein output[69,70]. In the nucleus, CPEB1 regulates alternative 3′ UTR formation[25]. Shortened 3′ UTRs have fewer miRNA and RNA-binding protein binding sites and therefore are less susceptible to post-transcriptional regulation. Intriguingly, we observed CPEB1 was localized in the cytoplasm of QSCs while it is expressed both in nuclei and cytoplasm of ASCs. Here, we mainly focus on the cytoplasmic functions of CPEB1 to regulate the SC quiescence-to-activation transition by reprogramming the translational landscape. A better understanding of the nuclear functions of CPEB1 in fully activated SCs is required to fully understand how CPEB1 regulates SC function.

Myod1 functions as an essential activator for SC activation. We showed that CPEB1 is a regulator of Myod1. Previous reports and our data both suggest that most SCs express Myod1 protein within hours of in vitro culture. However, it takes at least 16 hours for the siRNA to silence the gene expression effectively[55]. Therefore, the loss-of-function approach by siRNA is not ideal to study the regulation of Myod1 or SC early activation. Thus, we measured the effect of CPEB1 on Myod1

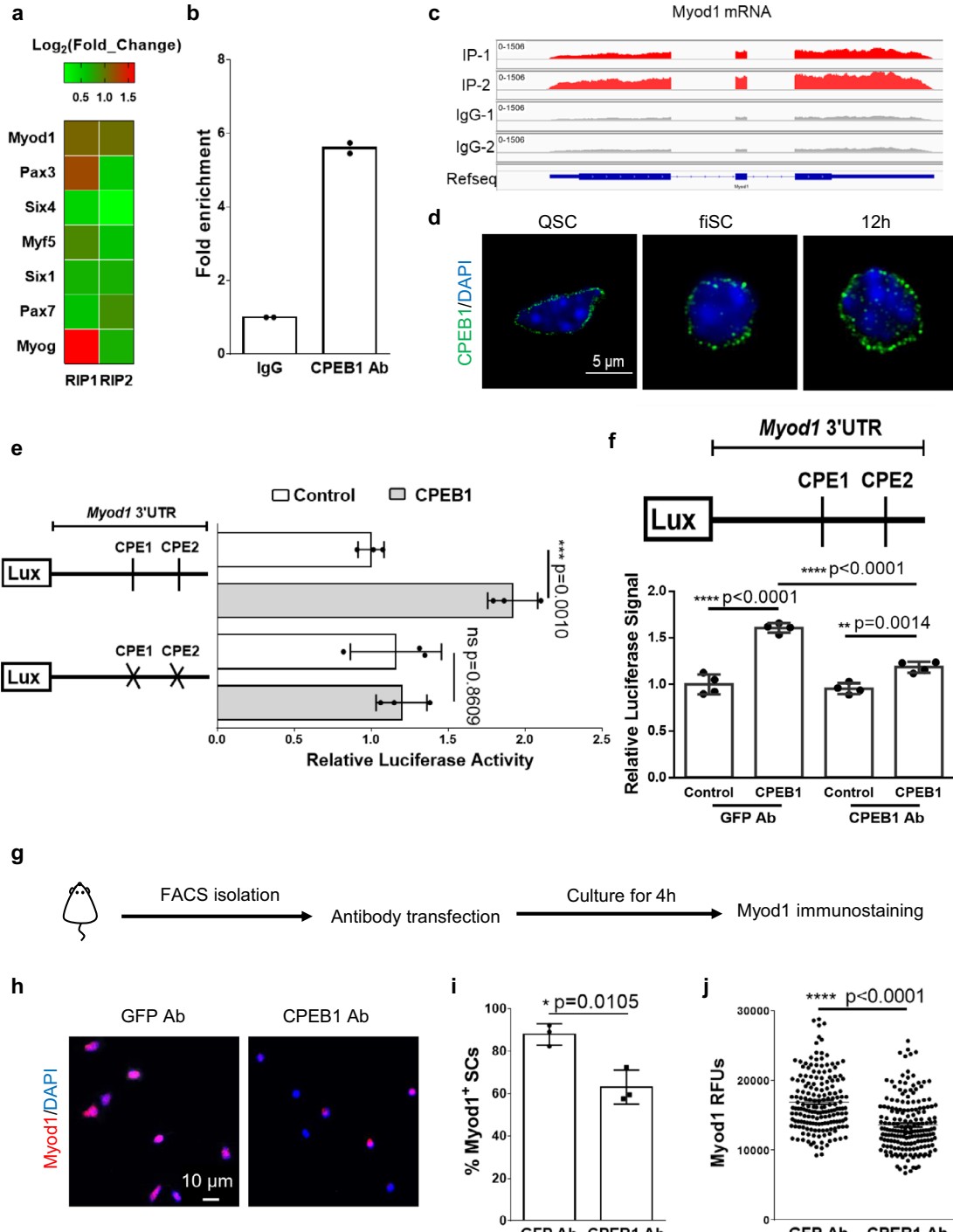

**Fig. 5 CPEB1 regulates Myod1 protein output by acting on the CPEs within its 3'UTR. a–c a** Heatmap of the Log$_2$(Fold_Change) of muscle factors in the CPEB1 RIP-Seq analysis. **b** Real-time PCR (qRT-PCR) analysis of *Myod1* mRNA after CPEB1 RNA-IP on fiSCs. ($n = 2$ independent experiments). **c** CPEB1 RIP-seq data of Myod1 transcripts. **d** CPEB1 immunostaining on QSCs, fiSCs and 4 hour cultured SCs (12 hours after mice were sacrificed). Nuclei were stained with DAPI. ($n = 3$ independent experiments). **e** *Myod1* 3'UTRs with wild-type (WT) or mutated cytoplasmic polyadenylation elements (CPEs) were inserted downstream of firefly luciferase (LUX) and co-transfected with pcDNA3.0-flag or pcDNA3.0-flag-CPEB1 into 293 T cells along with the Renilla luciferase vector as an internal control. Luciferase activity was quantified by dual-luciferase assay 36 hours after plasmid transfection. ($n = 3$ independent experiments). **f** The WT Myod1 3'UTR was inserted downstream of LUX and co-transfected with pcDNA3.0-flag or pcDNA3.0-flag-CPEB1 into 293 T cells along with the Renilla luciferase vector. One day after transfection, GFP or CPEB1 antibody (Ab) was transfected into 293 T cells. 4 hours after antibody transfection, the medium was replaced with fresh medium. One day afterwards, cells were harvested for dual-luciferase assay. ($n = 4$ independent experiments). **g–j** Myod1 protein expression analysis on SCs after CPEB1 antibody transfection. **g** Schematic illustration of Myod1 protein expression analysis on CPEB1 antibody transfected SCs. **h** GFP or CPEB1 antibody (Ab) was transfected into SCs. 4 hours after antibody transfection, SCs were subjected to Myod1 immunostaining. **i** Quantification of Myod1$^+$ SCs ($n = 3$ independent experiments) and **j** Myod1 RFUs after antibody transfection (the number of quantified cells are 177 and 186 for the GFP Ab and CPEB1-Ab groups, respectively). Data are presented as mean ± SD in panels **e**, **f**, **i**, **j**. The *p*-values calculated by two-tailed unpaired *t*-test are used for comparing two groups in **e**, **f**, **i**, **j** ns not significant. Source data are provided as a Source Data file.

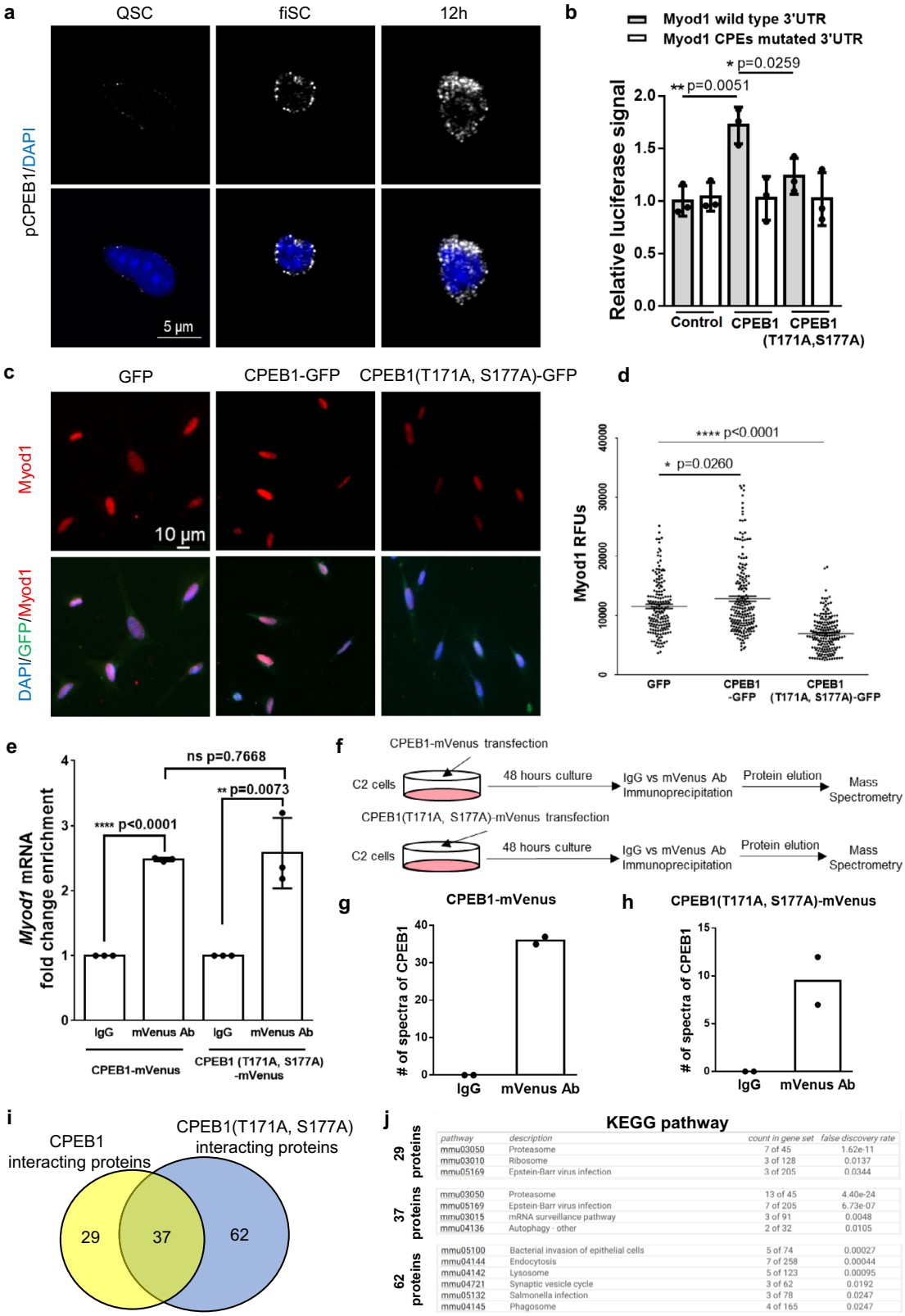

expression by CPEB1 antibody transfection, which neutralizes protein function rapidly[71–73]. Antibody transfection could therefore be more suited to study proteins and pathways that are important for the early activation of SCs.

In this study, we demonstrated that CPEB1 regulates the translational landscape during SC activation, (the working model is summarized in Supplementary Fig. 15). CPEB1 post-transcriptionally

regulates its targets by acting on the CPEs within the 3′ UTRs in a phosphorylation-dependent manner in transcripts such as those encoding for myogenic factor Myod1, metabolism, translation and signaling related proteins. We detected several ribosomal proteins in our CPEB1 immunoprecipitation experiment (Supplementary Data 5), and the interaction between CPEB1 and ribosomal proteins suggests a distinct angle for the regulation of CPEB1-mediated

**Fig. 6 Phosphorylation is required for CPEB1 to regulate translation. a** pCPEB1 immunostaining of QSCs, fiSCs and 4 hours cultured SCs (12 hours after mice were sacrificed). Nuclei were stained with DAPI. ($n = 3$ independent experiments). **b** The WT or CPE mutated 3′UTR of *Myod1* was inserted into the pmiR-report luciferase vector and co-transfected with pcDNA3.0-flag (Control), CPEB1, or CPEB1 (T171A, S177A) mutant into 293 T cells along with the Renilla vector as an internal control. 36 hours after transfection, luciferase activity was quantified by the dual-luciferase assay. ($n = 3$ independent experiments). **c, d** FACS-isolated SCs were infected with the indicated adenovirus for 36 hours. **c** Myod1 immunostaining on SCs infected with adenovirus-containing GFP, CPEB1, or CPEB1 (T171A, S177A). **d** Quantification of Myod1 RFUs on SCs after the indicated virus infection. ($n = 3$ independent experiments, the number of quantified cells is 166, 197, and 182 for the GFP, CPEB1, and CPEB1 (T171A, S177A) groups, respectively. **e** C2 cells were transfected with CPEB1-mVenus or CPEB1 (T171A, S177A)-mVenus. After 48 hours, C2 cells were harvested and used for mVenus RNA-IP. *Myod1* mRNA level was analyzed by qRT-PCR. ($n = 3$ independent experiments). **f–h** Proteomic analysis of CPEB1 interacting proteins. **f** Schematic illustration of the workflow for CPEB1-associated protein analysis. **g** The number of spectra of CPEB1 proteins detected in C2 cells transfected with CPEB1-mVenus or **h** CPEB1 (T171A, S177A)-mVenus after immunoprecipitation. ($n = 2$ independent experiments). **i** Venn diagram of CPEB1 and phosphorylation mutated CPEB1-specific interacting proteins. **j** Functional enrichment analysis of corresponding proteins subsets in **i**. Data are presented as mean ± SD in panels **b**, **d**, **e**. The *p*-values calculated by two-tailed unpaired *t*-test are used for comparing two groups in **b**, **d**, and **e**, ns not significant. Source data are provided as a Source Data file.

translation. Moreover, using CPEB1 RIP-seq, we identified the global targets of CPEB1, which are mainly enriched in translational regulations. The CPEB1-dependent proteome further consolidated the notion that CPEB1 reprograms the translational landscape by controlling translational regulators during SC activation. Together, this suggests that CPEB1 regulates the translation of different sets of transcripts to provide a proper cellular environment for SC activation.

In addition to its role in blood glucose homeostasis, insulin was shown to stimulate muscle protein anabolism and decrease muscle protein breakdown, leading to muscle hypertrophy. A local infusion of an intermediate dose of insulin in the physiological range resulted in increased blood flow and amino acid availability, leading to increased muscle protein synthesis with no change in muscle protein degradation[74]. However, a high dose neither affects blood flow nor protein synthesis, but reduces muscle protein breakdown, leading to improvements in muscle protein balance but to a lesser extent than the intermediate dose[74]. Similar studies confirmed the elevated muscle protein anabolism by physiological hyperinsulinemia[75,76]. The mechanism of elevated protein synthesis by insulin was associated to enhanced binding of eIF-4G to eIF-4E[77]. Muscle hypertrophy can be achieved by either increasing the RNA and protein output with the same number of nuclei or increasing the nuclei number[78]. As myonuclei are terminally differentiated and are unable to divide, the main contributor for fused nuclei is SCs. Early studies reported an increase in myonuclei number during synergistic ablation-induced hypertrophy in rats[79]. By utilizing SC-depleted mice and a synergistic ablation (SA) model, it was shown that the increase in myonuclei number is SC-dependent[7]. Other studies demonstrated neither inhibition of DNA replication or SCs ablation by irradiation affected the degree of hypertrophy[80,81].

While it is still debatable whether SCs are required for muscle hypertrophy, the role of insulin signaling in SC behavior is also not completely understood. Past studies suggested insulin promotes both myoblast proliferation and differentiation through the mTORC1 signaling pathway[82–87]. In this manuscript, we showed that insulin promotes SC activation and proliferation partially via the CPEB1 pathway. Our in vivo data also indicated that insulin improves muscle regeneration. Thus, our study may provide insights supporting a model where insulin regulates muscle hypertrophy by controlling SC proliferation via CPEB1. Our data showed that insulin regulates SC function. We acknowledge that owing to the correlative nature of the data, we cannot directly demonstrate that insulin stimulates phosphorylation of CPEB1 to regulate SC function. Since the insulin-mediated signaling pathway is complex, we cannot exclude the possibility that insulin regulates SC function by targeting other signaling pathways, such

as the mTORC1 pathway[82–87]. Here, our data may provide another possible insulin downstream target, CPEB1, for its regulation of cellular function. Of note, the pCPEB1 level was not rescued by insulin following MK5108 treatment and the negative effects of MK5108 on muscle regeneration and Pax7[+] SC number was not rescued by insulin (Fig. 7). A possible explanation is that MK5108 might cause severe cell cycle arrest because it not only inhibits CPEB1 phosphorylation but is also a highly selective inhibitor of Aurka, which is critical for centrosome organization during the cell cycle progression[88–90]. Furthermore, insulin does not directly phosphorylate CPEB1 but acts through a multi-step signaling pathway[59]. Thus, it is possible that the defects induced by MK5108 could not be rescued by insulin treatment, but the regeneration effects by MK5108 and insulin are appeared to act partially via CPEB1. Aurka is a Serine/Threonine protein kinase that can phosphorylate other proteins. We cannot exclude the possibility that the effects on SCs by MK5108 might act through alternative signaling pathways such as PI3-K/Akt, mTOR, β-catenin/Wnt and NF-κB pathways[91]. Taken together, our data in this study suggests CPEB1 may be a potential Aurka downstream target, regulating SC function with regards to stem cell self-renewal and muscle regeneration.

## Methods

**Mouse lines**. C57BL/6 and Pax7-nGFP mice were maintained at the Laboratory Animal Facility at the Hong Kong University of Science and Technology (HKUST). Mouse experiments were performed using males in accordance with the criteria of the Special Health Services, Department of Health, HKSAR Government. The mice were maintained in an animal room with 12 h light/12 h dark cycles, room temperature kept at 22–24 °C, with humidity of 40–60%.

**Mice anesthesia and euthanasia**. The mice were anesthetized by intraperitoneal (IP) injection of tribromoethanol (Avertin) at a dosage 500 mg/kg. Mice were sacrificed by Carbon dioxide ($CO_2$) inhalation followed by cervical dislocation.

**Satellite cell isolation and culture**. Satellite cells (SCs) were isolated as described[60]. Briefly, hindlimb muscles were dissected from mice and minced in dissociation buffer (900U/ml Collagenase II (Gibco, cat#17101015) in Ham's F-10 medium (Sigma–Aldrich, cat#N6635) containing 10% horse serum (HS, Invitrogen, cat#16050114) and 1% penicillin/streptomycin (P/S, Thermo Scientific, cat#15140122). After two rounds of enzyme digestion, the single-cell suspension was passed through a 40 μm cell strainer (SPL, cat#93040). For B6 (C57BL/6) mice, the cell suspension was stained with FITC anti-CD31 (BioLegend, cat#102506), FITC anti-CD45 (BioLegend, cat#553080), Alexa 647 anti-Sca1 (BioLegend, Cat#108118), Biotin anti-mVcam1 (BioLegend, Cat#105704) antibodies. The Biotin anti-mVcam1 was stained with streptavidin PE-Cy7 antibody (BioLegend, Cat#405206). The stained single-cell suspension was analyzed and sorted by FACS using the BD Influx flow cytometer. The CD31[−], CD45[−], Sca1[−], mVcam1[+] population was sorted as SCs. FACS-sorted SCs were then cultured in Ham's F-10 medium containing 10% HS and 1% P/S. Without further indications, SCs were sorted from C57BL/6 mice. In this manuscript, freshly isolated SCs were termed as fiSCs. The fixed-satellite cell isolation procedure followed the protocol as described[42,13]. Briefly, Pax7-nGFP mice were perfused with 30 mL of 1× (phosphate-buffered saline) PBS followed by a 30 mL of 0.5% paraformaldehyde (PFA)

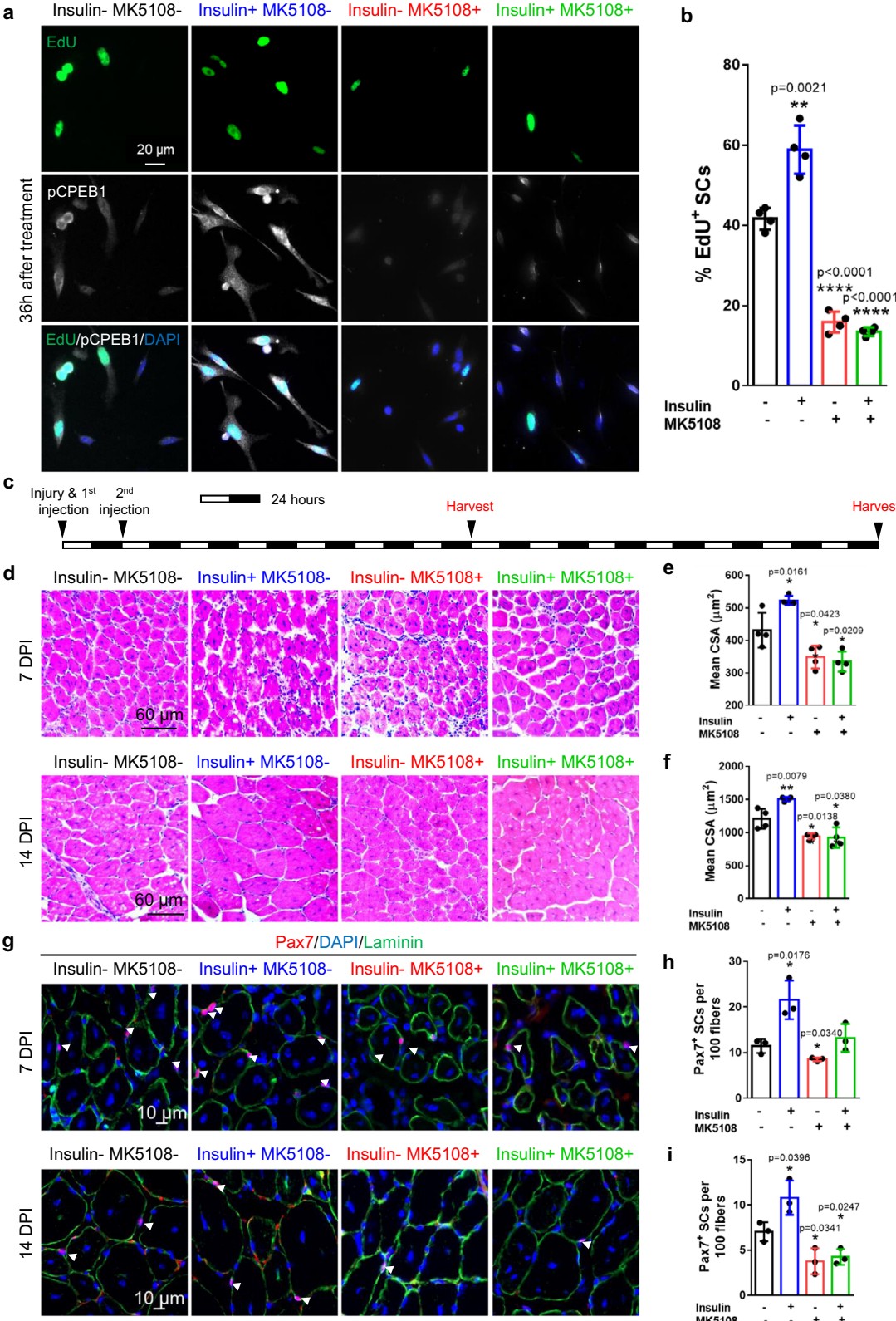

fixation for 5 min. PFA was subsequently quenched by 30 mL of 2 M glycine (Sigma–Aldrich, cat#G8898) perfusion for 5 min. In this manuscript, the SCs sorted from PFA-perfused muscle were termed as quiescent SCs (QSCs).

**Single muscle fiber extraction.** Extensor digitorum longus (EDL) muscles were dissected and digested in Ham's F-10 medium with 800 U/mL collagenase II for 85 min in a shaking water bath at 37 °C. To extract fibers from perfused mice, the EDL muscles were digested with 1600 U/mL collagenase II. The fibers were then triturated, washed extensively in Ham's F-10, 10% HS with 1% P/S, and cultured in Ham's F-10 medium containing 20% fetal bovine serum (FBS, Gibco, cat#10270-106), 0.01% basic fibroblast growth factor (bFGF, PeproTech, cat#100-18B) and 1% P/S. Fibers were cultured in suspension, and half of the medium was refreshed every day. Without further indications, fibers were extracted from C57BL/6 mice.

**Cell lines.** Both mouse C2C12 myoblast (CRL-1772) and 293 T cell lines (CRL-3216) were obtained from the American Type Culture Collection (ATCC) and

**Fig. 7 Pharmacological manipulation of CPEB1 phosphorylation regulates SC activation, muscle regeneration and Pax7$^+$ SC number after regeneration.**
**a**, **b** FACS-isolated SCs were plated down and treated with MK5108 and/or insulin for 36 hours. Cells were also continuously supplied with EdU for EdU incorporation analysis. **a** 36 hours after MK5108 and/or insulin treatment, SCs were harvested for EdU detection and pCPEB1 immunostaining. Nuclei were stained with DAPI. **b** Quantification of EdU$^+$ SCs after MK5108 and/or insulin treatment. ($n = 4$ independent experiments). **c** Timeline of muscle injury, intramuscular injection of MK5108 and/or insulin, and muscle regeneration study for **d**–**i**. **d**–**f** Tibialis anterior (TA) muscles were injured and injected with insulin and/or MK5108, then allowed to regenerate. **d** Histological analysis of the cross-sectioned TA muscles using hematoxylin and eosin (H&E) staining 7 days and 14-days post-injury (DPI). **e**, **f** Quantification of the size (in cross-sectional area) of regenerated fibers in **d**. ($n = 4$ independent experiments). **g**–**i** TA muscles were injured and injected with insulin and/or MK5108, then allowed to regenerate. **g** Immunostaining for Pax7 and laminin of cross-sectioned muscle fibers 7DPI and 14DPI. **h**, **i** Quantification of Pax7$^+$ SCs in **g**. ($n = 3$ independent experiments). Data are presented as mean ± SD in panels **b**, **e**, **f**, **h**, **i**. The $p$ values calculated by two-tailed unpaired $t$-test are used for comparing two groups in **b**, **e**, **f**, **h**, and **i**, ns not significant. Source data are provided as a Source Data file.

cultured in DMEM (Gibco) medium with 10% FBS (Gibco), 100 units/ml of penicillin, and 100 μg of streptomycin (P/S, Thermo Scientific, cat#15140122) at 37 °C in 5% CO$_2$. Wild-type primary myoblast cells were cultured in Ham's F-10 medium supplemented with 20% FBS and bFGF (0.025 mg/ml). For the in vitro CPEB1 siRNA efficiency assay, primary myoblasts were plated and cultured to 30% confluence followed by siRNA transfection. Two days post-transfection, the cells were harvested for qRT-PCR and western blotting. For the CPEB1 or CPEB1 (T171A, S177A) interacting proteins experiments, C2C12 cells were plated into 10 cm plates and cultured to 30% confluence followed by the CPEB1-mVenus or CPEB1 (T171A, S177A)-mVenus plasmid transfection. After two days of culture, C2C12 cells were harvested for mVenus antibody immunoprecipitation followed by mass spectrometry. For the dual-luciferase assay, 293 T cells were plated into 96-well plates and cultured to 50% confluence followed by luciferase vectors transfection. After 36 h of culture, 293 T cells were harvested for luciferase signal detection. For the CPEB1 siRNA specificity experiments, 293 T cells were plated into 6-well plates and cultured to 30% confluence followed by pcDNA3.0-CPEB1 plasmid transfection. One day post-transfection, CPEB1 siRNA was transfected into the cells. After two days in culture, 293 T cells were harvested for qRT-PCR and western blotting.

**Muscle injury**. Before the injury, the mice were anesthetized by IP injection of tribromoethanol. The TA muscles were injured by injecting 30 μL of 1.2% Barium Chloride (BaCl$_2$). Shank muscles were injured for injured activated SC sorting by evenly injecting 50 μL of 1.2% BaCl$_2$, followed by even stabbing using a 31 G insulin syringe (BD, cat#328440) ~50 times.

**Immunofluorescence**. Fibers or cells were fixed with 4% PFA and permeabilized by 0.1% PBS with Tween 20 (PBST), followed by blocking in 5% goat serum (ThermoFisher Scientific) (diluted in 0.1% PBST) and incubated with primary antibodies overnight at 4 °C. After washing three times in 0.1% PBST, the cells/fibers were incubated with secondary antibodies and 4′,6-diamidino-2-phenylindole (DAPI) for 30 min at room temperature. Cells/fibers were subsequently washed 3 times with 0.1% PBST for 30 min each time, followed by slide mounting with Fluoro-Gel (Electron Microscopy Sciences, cat#17985-11).

**siRNA knockdown**. For 1 mL of culture medium per well of a 6-well plate, 50 μL of Opti-MEM (Sigma–Aldrich) and 1 μL of Lipofectamine 3000 (Invitrogen) were combined with 50 μL of Opti-MEM and 2.5 μL of siRNA (20 μM). This cocktail was incubated for 15 min at room temperature and then added to the cultured cells or fibers. 8 h after siRNA transfection, the medium was replaced with a fresh pre-warmed medium. The siCPEB1 and negative control siRNA were both obtained from Ribobio.

**siRNA specificity test**. 293 T cells were plated for 1 day in 6-well plates followed by transfection with wild-type (WT) or mutant CREB1 containing constructs. One day later, siRNA targeting either WT or mutant CPEB1 was transfected into the cells. Two days following the second transfection, the cells were then harvested for CPEB1 mRNA or protein detection.

**Molecular cloning**. The mouse CPEB1 gene was cloned into the pcDNA3.0-mVenus plasmid using Gibson assembly (NEB, cat#E2611S) following the manufacturer's instructions. The CPEB1 T171A, S177A or T171D, S177D phosphorylation mutant was generated using the QuikChange II Site-Directed Mutagenesis Kit (Stratagene) according to the manufacturer's instructions.

**Histology**. Tibialis anterior (TA) muscles were dissected and fixed in 0.5% PFA for 6 h, followed by dehydration in 20% sucrose (Scharlau) solution overnight at 4 °C. On the following day, TA muscles were immersed in Optimal Cutting Temperature (O.C.T.) (Sakura) compound in a base mold (Electron Microscopy Sciences) and then frozen in isopentane (Sigma–Aldrich) pre-chilled using liquid nitrogen. TA muscles were sectioned at 6 μm thickness using a cryostat (NX70, Thermo Fisher

Scientific). For hematoxylin and eosin (H&E) staining, cryo-sectioned muscles were stained with hematoxylin for 10 min, washed with Scott's water for 5 min, and stained with eosin for 2 min. Muscles were subsequently dehydrated in 95% ethanol followed by 100% ethanol and cleared twice with xylene. Slides were mounted with DPX medium (Sigma–Aldrich). For immunofluorescence staining of uninjured TA muscles or 6 days post-injured muscles, sections were rehydrated in 1× PBS followed by fixation with 4% PFA for 5 min. Sectioned muscles were subsequently blocked with unconjugated anti-mouse Fab (1:50, Jackson ImmunoResearch) diluted in 0.3% PBST for 2 h. Primary antibodies were added to the slides and incubated overnight at 4 °C. Before incubation with secondary antibodies, the slides were washed three times with 0.3% PBST. After incubation with anti-Fab secondary antibodies and DAPI at room temperature for 15 min, the slides were washed three times with 0.3% PBST and mounted. For the immunostaining of TA muscles 3 days post-injury, fixation and blocking were performed in the same manner as for uninjured TA muscles. Next, sectioned muscles were stained with direct-conjugated antibody for Myod1 (generated using the APEX antibody labeling kit; Thermo Fisher Scientific) overnight at 4 °C. Other protein immunostaining followed the primary antibodies and secondary antibodies staining protocol as described above. Briefly, the cells were fixed in 4% PFA followed by blocking in 5% goat serum. After incubation with primary antibody overnight at 4 °C, the slides were washed three times for 30 min. Secondary antibodies and DAPI staining were performed for 15 min at room temperature. After washing away the unbound secondary antibodies, the slides were mounted.

**Real-time PCR**. Total RNA was isolated using the NucleoSpin RNA XS kit (Macherey-Nagel) according to the manufacturer's instructions. Isolated RNA was quantified using the Qubit (Thermo Fisher Scientific). cDNA was generated using the High-Capacity cDNA Reverse Transcription Kit (Invitrogen). Real-time PCR was performed in a 10 μL reaction volume containing 5 μL of 2× SYBR Master Mix (Roche), 4 μL of cDNA template, and 1 μL of 375 nM primer. Real-time PCR was performed on the Light-Cycler 480 (Roche).

**Western blot analysis**. Cultured cells were washed with pre-chilled 1× PBS and lysed in RIPA buffer supplemented with cOmplete Protease Inhibitor Cocktail (Sigma–Aldrich), and subsequently scraped off the dish and transferred to a pre-chilled tube. The lysates were then centrifuged at 12,396 × $g$ (Eppendorf, Centrifuge 5418) for 20 min at 4 °C. The supernatant was collected, quantified using the Pierce BCA Protein Assay Kit (Thermo Fisher Scientific), and boiled with 6× sample loading buffer for 5 min at 95 °C. The lysate was electrophoresed on a 10% polyacrylamide gel followed by nitrocellulose membrane (Bio-Rad) transfer. The membrane was blocked in 5% non-fat milk dissolved in 0.05% TBST for 30 min in a rocker with agitation at room temperature, and then subsequently incubated with primary antibodies overnight at 4 °C and washed three times with 0.05% TBST. The membrane was then incubated with secondary antibodies for 30 min at room temperature, washed again three times with 0.05% TBST, and analyzed using the Odyssey imaging system (LI-COR).

**Dual-luciferase assay**. CPEB1 was cloned into the pcDNA3.0-flag vector (Addgene), and the *Myod1* 3′ untranslated region (UTR) was cloned into the pmiR vector (Promega). Cytoplasmic polyadenylation element (CPE)-mutated *Myod1* 3′-UTR generated by High-Fidelity KAPA HIFI Hot Start 2× Ready Mix (Kapa Biosystems) was cloned into the pmiR vector. Next, 20 ng of pmiR-*Myod1* 3′-UTR, 100 ng of pcDNA3.0-flag or pcDNA3.0-flag-CPEB1, and 5 ng of pRL-TK were co-transfected into 293 T cells cultured in 96-well plates. Using the Dual-Luciferase Reporter Assay System Kit (Promega), luciferase activity was measured on the Lumat LB9507 luminometer (Berthold Technologies).

**5-ethynyl-2′-deoxyuridine (EdU) incorporation and detection**. EdU was added to the cultured FACS-sorted cells or fibers at a final concentration of 10 μM. The EdU signal was visualized using the Click-iT EdU Imaging Kit (Invitrogen) according to the manufacturer's instructions.

**Single-molecule fluorescence in situ hybridization (smFISH)**. *Myod1* transcripts probes were obtained from Stellaris, and smFISH for *Myod1* transcripts was performed according to the manufacturer's protocol.

**MK5108 and insulin treatment**. MK5108 (32 μM) or insulin (5 μg/mL) was added to cultured SCs. For MK5108 or insulin intramuscular injection, 0.037 mg of MK5108 or 0.01 mg of insulin was injected into each TA muscle of 8-week-old mice immediately after injury. The drug was injected once more on the next day after injury. TA muscles were then dissected 7- or 14-days post-injury.

**CPEB1 RNA immunoprecipitation sequencing and data analysis**. 1,000,000 freshly isolated SCs were harvested in polysome lysis buffer (PLB)[92] with protease inhibitor (Sigma) and RNase inhibitor (Invitrogen). The lysate was precleared using Protein A Dynabeads (Thermo Scientific). CPEB1 antibody (10 μg) (Abcam) was added to the precleared lysate and incubated with rotation at 4 °C overnight. Protein A Dynabeads were resuspended using PLB and added to the SC lysate and rotated at 4 °C for 4 h. The lysate was put on a magnetic stand and the supernatant discarded. Beads were washed with PLB for four times, rotated at 4 °C for 5 min per wash. RNA was isolated using the Nucleospin RNA XS kit and the cDNA library was generated using the SMART-Seq2 method followed by sequencing on the Illumina Nextseq 500 using 2 × 75 kit. Data analysis was performed following the published protocol[93]. Briefly, the raw reads were mapped to mm10 genome using HISAT 2.1.0. The Sam files were sorted and converted to Bam files by Samtools-1.9. Reads were assembled and quantified by Featurecounts-1.5.1, and then the fragments per kilobase of transcript per million mapped reads (FPKM) were calculated. We then normalized the FPKM of genes of the IP sample to the IgG sample of each experiment. The raw read counts were used to perform DEseq2-1.30.1[94] analysis to determine the CPEB1 antibody-enriched genes. To exclude potential false-positive results, we set a FPKM fold-change of IP/IgG of each experiment >2 and the P-adj < 0.05. The scatter plot of CPEB1 binding genes was generated using GraphPad 7. Gene ontology (GO) analysis was performed using g:Profiler[95].

**Adenovirus packaging and infection**. To generate adenovirus, 90% confluent 293 A packaging cells in a 6-well plate were transfected with 1 μg of pAd/BLOCKiT-DEST vectors carrying GFP or CPEB1-GFP using Lipofectamine 2000. After transfection, viruses were generated using the Block-iT Adenoviral RNAi Expression System (Invitrogen) according to the manufacturer's instructions. After 12 days, the viral supernatant was harvested for subsequent amplification and concentration. The titer of adenovirus was approximately $1 \times 10^7$ pfu/ml. SCs were infected with the adenovirus-containing medium for 12 h.

**Protein transfection**. GFP antibody or CPEB1 antibody was transfected into SCs using the X-fect Protein Transfection Kit (Clontech) according to the manufacturer's instructions.

**λ protein phosphatase treatment**. Primary myoblasts were cultured in 10 cm plates and lysed in RIPA buffer with EDTA-free protease inhibitor cocktail (Sigma–Aldrich). Total protein was subjected to λ protein phosphatase (NEB) treatment following the protocol from NEB. Afterwards, western blotting was performed to detect pCPEB1 (anti-pCPEB1 diluted at 1:200) using the protocol as described above.

**Co-analysis of transcriptome and proteome**. The FPKM table of QSCs and ASCs RNA-seq was previously generated in our lab. The fold change between samples was calculated based on the FPKM values (transcriptome) and NSAF (proteome), followed by a Student's *t*-test. The fold change was log2 normalized.

**Antibodies**. The antibodies used for immunostaining in this study were Myod1 (Dako, Clone 5.8 A,1:500), Ki67 (Abcam, ab16667, 1:100), Pax7 (DSHB, 1:50), Laminin (Abcam, ab11576, 1:1,000), Syn4 (kind gift from Prof. Dawn Cornelison, 1:500), CPEB1 (Abcam, Ab73287, 1:500), pCPEB1 (China-peptides, 1:200), Aurka (Proteintech Group, 10297-1-AP, 1:200), pAurka (Thermo-Scientific, 44-1210 G, 1:500), GFP antibody (Invitrogen, A11122, 10 μg for immunoprecipitation), and Alexa Fluor 488, 594, 647 donkey anti-mouse, anti-rabbit, anti-rat (Invitrogen, A2102, A2103, A31571, A21206, A21207, A31573, 1:1,000). Anti-mouse Fab (1:50, Jackson ImmunoResearch, lot:153348), Myogenin (BD, 1:500. cat: 556358, Clone: F5D).

**Enzyme-linked immunosorbent assay (ELISA)**. ELISA microplates (Sigma–Aldrich) were coated with phosphor or non-phosphor peptides (1 μg/ml) overnight at 4 °C. Plates were then blocked with 5% milk at 37 °C for 1.5 h. Rabbit IgG or pCPEB1 antibody at different dilutions was added to each well and incubated at 37 °C for 1 h. This was followed by washing with Milli-Q water ten times. A secondary antibody (Goat anti-Rabbit, 1:10,000 dilution, Abcam) was added to each well and incubated for 30 min. This was followed by washing with Milli-Q water ten times. The chromogenic substrate TMB (Sigma–Aldrich) was then added to each well and the microplate incubated for 15 min in the absence of light. The

reaction was stopped by adding sulfuric acid (2 M). The OD450 was measured using a spectrophotometer (APL).

**Mass spectrometry and label-free quantification of mass spectrometry data**. Proteins from SCs were extracted using RIPA buffer. For the prefixed QSCs, the protein lysate was heated at 70 °C for 2 h to de-crosslink the fixed protein. The protein was precipitated by adding 4× volume of pre-cooled acetone. The protein pellet was washed with pre-cooled acetone, followed by pre-cooled ethanol and then pre-cooled acetone. The pellet was resuspended with UA buffer (8 M urea in 0.1 M Tris.HCl) followed by the addition of dithiothreitol (DTT, final concentration 2 mM) and incubated at 30 °C for 1.5 h. Iodoacetamide (IAA) (Sigma–Aldrich) (final concentration 10 mM) was added to the sample and incubated protected from light for 40 min. Afterwards, trypsin was added (final concentration 0.25 μg/μl) into the suspension for overnight digestion. The digestion was stopped by adding trifluoroacetic (TFA) (Sigma–Aldrich) (final concentration 0.4%). After salt depletion using C18 spin tips (Thermo Scientific), the material was loaded onto the Bruker timsTOF Pro mass spectrometer by following the manufacturers of Bruker. For Liquid chromatography (LC), mobile phase A is 98% MilliQ Water, 2% Acetonitrile with 0.1% Formic Acid and mobile phase B is 100% Acetonitrile with 0.1% Formic acid. We used the ionoptiks 25 cm Aurora Series emitter column with CSI (25 cm × 75 μm ID, 1.6 μm C18). For a detailed description of the mass spectrometer, please see ref. [96] Briefly, the Captive Spray Ion source provides the ions which enter the first vacuum stage and accumulate in the front part of the dual Trapped Ion Mobility Spectrometry (TIMS) analyzer. The ion cloud is confined by the 300 Vpp RF potential radially. After the initial accumulation step, ions are transferred to the second part of the TIMS analyzer for ion-mobility analysis. In both parts of the TIMS analyzer, the RF voltage is superimposed by an electrical field gradient (EFG). Thus, the ions in the tunnel are pulled by the incoming gas flow from the source and simultaneously retained by the EFG. Afterwards, the ions are released from the TIMS analyzer in order of their ion mobility for QTOF mass analysis. The dual TIMS setup allows operation of the system at 100% duty cycle, when accumulation and ramp times are kept equivalent. The accumulation and ramp time are set at 100 ms each and mass spectra are recorded in the range from the ratio of *m/z* 100–1700 using the positive electrospray mode. The ion mobility was scanned from 0.85 to 1.30 Vs/cm2. The overall acquisition cycle of 0.53 s comprised one full TIMS-MS scan and four Parallel Accumulation-Serial Fragmentation (PASEF) MS/ MS scans. The TIMS dimension was calibrated linearly using three selected ions from the Agilent ESI LC/MS tuning mix [*m/z*, 1/K0: (622.0289, 0.9848 Vs cm$^{-2}$), (922.0097, 1.1895 Vs cm$^{-2}$), (1221,9906, 1.3820 Vs cm$^{-2}$)] in positive mode.

The raw data was processed by PEAKS software (Version: X + ). The database for searching the proteomic data is Uniprot, the taxonomy is Mus musculus. The parent ion is 15ppm. The fragment ion is 0.05 Da. The protein FDR is 1%. We applied spectral counting for the label-free quantification followed by the normalized spectral abundance factor (NSAF) method[97]. Simply, we obtained the spectral abundance factor by normalizing the spectral number of proteins with the length of the protein. We then normalized the spectral abundance factor between samples by dividing by the sum of all the spectral abundance factors. Since the NSAF was very small, we multiplied the NSAF by 10$^6$. The Student's *t*-test was performed to calculate the differentially expressed proteins between samples. We used one-pair distribution and homoscedastic Student's *t*-test. The level of statistical significance was set at *p* < 0.05. Hierarchical clustering analysis of the protein expression was performed by Biovinci 1.1.5.

**Preparation of C2C12 cells after mVenus antibody immunoprecipitation for mass spectrometry and qRT-PCR analysis**. C2C12 cells were seeded into 10 cm tissue culture plates and transfected with 20 μg of CPEB1-mVenus or CPEB1 (T171A, S177A)-mVenus plasmid using ViaFect™ Transfection Reagent (Promega). After two days of culture, C2C12 cells were harvested in PLB[92] for mVenus antibody immunoprecipitation following the steps mentioned above in CPEB1 immunoprecipitation. Briefly, the cells were lysed in PLB buffer, and the lysate was precleared using Protein A Dynabeads. GFP antibody (10 μg) (Invitrogen, A11122) was added to the precleared lysate and incubated with rotation at 4 °C overnight. Protein A Dynabeads were resuspended using PLB and added to the cell lysate and rotated at 4 °C for 4 h. The lysate was put on a magnetic stand and the supernatant discarded. Beads were washed with PLB for four times, rotated at 4 °C for 5 min per wash. The RNA was isolated, followed by qRT-PCR of *Myod1* mRNA. Protein was eluted by adding 100 μl of RIPA buffer to the beads for 10 min at 95 °C. The protein was precipitated with acetone from the supernatant for mass spectrometry analysis. By comparing with IgG control samples, we identified 66 proteins as WT CPEB1 interacting partners and 99 proteins as CPEB1 (T171A, S177A) interacting partners. To reduce the number of false-positive results, we only considered proteins detected in mVenus antibody IP groups but not in IgG control groups. To further determine the specific proteins interacting with WT CPEB1 or mutated CPEB1, we overlapped the two lists and identified the specific protein partners.

**Statistics and reproducibility**. All statistical analyses were performed using GraphPad Prism 7. Error bars in the figures represent the standard deviation (SD). The statistical significance was assessed by the Student's two-tailed paired and

unpaired *t*-test. ns indicates not significant. The level of statistical significance was set at $p < 0.05$. An ANOVA test was used to analyze multiple comparisons with the analysis result shown in the Source Data file. All raw data and analysis summaries are provided in the Source Data file. Representative images of at least three independent experiments are shown in Figures. Immunofluorescence images were quantified and analyzed by Zeiss ZEN Lite 2.5. We drew the cell boundary using the "Draw Spline Contour" tool of ZEN Lite 2.5. In the "Measure" window, all the information of the "drew cell" are provided. We used "Mean" as the relative fluorescence unit in this manuscript.

**Reporting summary**. Further information on research design is available in the Nature Research Reporting Summary linked to this article.

## Data availability

The CPEB1 RIP-seq data generated in this study have been deposited into the NCBI Gene Expression Omnibus database under accession code GSE148912. The proteomics data generated in this study have been deposited into the ProteomeXchange Consortium via the PRIDE[98] partner repository under accession code PXD018865, PXD020822, and PXD023656. Source data are provided with this paper.

## Code availability

For the code for CPEB1 RIP-seq analysis, please refer to the published protocol[93].

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

## Acknowledgements

We thank Y.C. Law and L.P. Ngan for their technical assistance and all members of Cheung laboratory for discussions. We also thank Prof. S. Tajbakhsh for providing us with the Pax7-nGFP mice. This work was supported by research grants from the Hong Kong Research Grant Council (GRF16102319, GRF16102420, C6018-19G, C6027-19G, AoE/M-604/16, T13-605/18 W), Lee Hysan Foundation (LHF17SC01), Hong Kong Epigenome Project (Lo Ka Chung Charitable Foundation), and the Croucher Innovation Award (CIA14SC04) from Croucher Foundation. This study was supported in part by the Innovation and Technology Commission (ITCPD/17-9). L.Y. is a recipient of the Hong Kong Ph.D. Fellowship Scheme. T.H.C. is the S H Ho Associate Professor of Life Science at HKUST.

## Author contributions

W.S.Z. and T.H.C. conceptualized the study and designed the experiments; W.S.Z., L.Y., K.S.W., and W.X.Z. carried out the experiments; T.H.C. provided supervision and guidance; W.S.Z., L.Y., W.K.S., E.H.Y.T., and T.H.C. drafted the manuscript.

## Competing interests

The authors declare no competing interests.
