## [Peer Review File · Nature Communications]

CPEB1 directs muscle stem cell activation by reprogramming the translational landscapeREVIEWER COMMENTS

Reviewer #1 (Remarks to the Author):

This manuscript from Dr. Cheung's lab describes the proteome of quiescent MuSCs and at early states of activation. The data shows a discordance between transcriptome and proteome, thus evidence for post transcriptional regulation of MuSC activation. The authors focus on CPEB1-a molecule with no known function in MuSCs. While CPEB1 regulates a significant part of the proteome the authors focus on CPEB1 role in Myod1 regulation.

This is a extremely novel study, the experiments are performed to highest standard and the data has uncovered a new layer of regulation for MuSC activation.

Major Strengths:

- 1) Provides a comprehensive characterization of proteins expressed in MuSCs during quiescence to activation transition.
- 2) Comparative analysis with transcriptome.
- 3) Both are a significant addition to the field and will provide scientists opportunity for many hypothesis forming questions.
- 4) The authors have identified CPEB1-this has no known function in MuSC regulation to date.
- 5) The authors go on to identify the mechanism of CPEB1-using both unbiased approach and then focusing onto Myod1. The mechanistic aspect of this study is comprehensive and conclusive.
- 6) Use of a blocking antibody to disrupt intracellular protein activity is a novel approach.
- 7) The use of in vivo, ex vivo and in vitro approaches are essential to provide the in vivo and mechanistic studies.

While there are always more experiments, a reviewer can ask. There are only a couple of issues that should be addressed- based on the authors conclusions.

After siRNA CPEB1 the authors report that MuSCs don't activate based on a reduction in fraction of MuSCs that enter S-Phase (in Figure 2 and S3), I would like to know whether these cells are stalled in G0, G1 or dying though apoptosis.

In the text, when discussing figure 7, please state in the text that insulin and MK518 was used to perturb muscle regeneration in vivo-for ease of reader.

The authors state that CPEB1 is required for a proper MuSC proliferation environment. The use of environment could be misleading.

Reviewer #2 (Remarks to the Author):

In the present article entitled "CPEB1 directs muscle stem cell activation by reprogramming the translational landscape" Zeng et al. investigate in vivo the translational and proteomic fingerprints of muscle stem cells using a novel whole body perfusion fixation technique able to obtain the unmodified quiescent signature avoiding common sample processing modifications . The manuscript is well-written, figures are nice and very illustrative, and the results are very interesting. Nonetheless, my main concern is the lack of proper discussion of results in discussion section, as this current version resembles a results section, specially the first parts. Thus, before publication in Nature Communications, I suggest doing some edits. Here, I detail my suggestions (or questions) point-by-point:

Abstract

- If muscle stem cells (MuSCs) and satellite cells (SC) correspond to the same subset of cells, then better use only one of the terms. Maybe authors could say: "In skeletal muscle, the muscle stem cells, also known as satellite cells (SC), are actively regulated in the quiescent state." And then only use one of the abbreviated terms. Now, both terms are used to refer to the same type of cells.

- In the sentence: "Using a whole mouse perfusion fixation approach,..." I would add: "Using a whole mouse perfusion fixation approach to obtain true quiescent SCs,..." otherwise, it was not clear in the abstract the objective of the whole mouse perfusion fixation approach.

- In a related manner, in the abstract authors should give more importance to the technique used in this study. Please include a sentence explaining what is able to achieve the whole body perfusion fixation technique to investigate SC cells. Currently it is not clear until the reader finished the introduction section.

Results:

Line 124 – Delete/substitute "dramatic". If authors want to add an adjective referring the observed changes maybe "significant" would be more adequate. I think dramatic is present in other parts of the text too.

-The full name of the abbreviation PFA is missing.

Discussion:

As I already mentioned, authors need to discuss more the findings obtained in this study with the existent studies published by other authors in discussion section. I suggest editing discussion section expanding it. For example the whole first page of discussion looks like results. Additionally, in this section is also missing a discussion about the methodology used, and the existent methodology used to study the quiescence-to-activation transition. If this is the first time that is done a study of this biological phenomenon using the whole body perfusion fixation this has to be clear in discussion.

Figures:

Figure 1a. I suggest editing a bit the diagram of section 1a. It is not very clear. Specify clearly what population of cells was obtained by FACS.

Methods:

- It is not clear in this section what is the procedure used by the authors to obtain every subset of cells used in this study. For example, in the materials and methods it is not mentioned the term fiSCs. Are these cells obtained without perfusion? Please ensure that all procedures are properly described in detail. The reader must be able to repeat all the experiments performed in this study reading the materials and methods.

- How many animals were used in the study? What is the n for the different experiments? Please detail it clearly in the text, in every experiment.

- Information about anesthesia (ej. Dose) and strategy of sacrifice need to be specified in the first section of methods.

- In "Satellite cell isolation and culture", I suggest to include a Briefly, ... and explain a little bit the procedure used to isolate SCs.

- As fixation by perfusion is the main technique used in this study I suggest to include an independent section in materials explaining the procedure used. Please locate it at the beginning of the section after the animals section to make the distribution linear for the reader.

- Please include PEAKS software version.

- In the section "Mass spectrometry and label-free quantification of mass spectrometry data." More information about the parameters used in Tims TOF instrument are missing. Authors should provide enough details to allow readers to repeat the analysis.

- Similarly, no information about the database search for proteomics data in PEAKS was detailed. Please mention the database used, the tolerance error for the parent ion and for fragments, as well as the FDR.

- I suggest editing the title "Mass Spectrometry and qRT-PCR on C2 cells after mVenus antibody immunoprecipitation." to something like "Preparation of C2 cells after mVenus antibody immunoprecipitation".

Refs.

I could not find ref 20 cited in the text.

- I would also suggest to include a conclusions section,

Reviewer #3 (Remarks to the Author):

The work by Zeng et al approaches the contribution of CPEB1 to translational regulation of muscle SC activation.

Following previous transcriptomic studies from the group (in QSCs, fiSCs and activated SCs), now the authors perform proteomic analyses in the same 3 cell types. In addition to the information pertaining the proteome changes during mSC activation, comparison of mRNA levels and proteome provides an indirect measure of potential translation regulation (and or Protein stability, see below). Following this observation, the authors focus in CPEB1 finding that this protein is overexpressed upon mSC activation (proliferation). Depletion of CPEB1 (siCPEB1) is associated with lower proliferation (cell division). Then the authors perform CPEB1-RIP in fiSC and proteomics on siCPEB1 SC. The authors identify over five hundred proteins affected in the CPEB1KD. Crossing this list with the CPEB1 RIP (With all the limitations noted below) the authors identify 39 CPEB1 regulated proteins in ASCs (of which 9 are related to translation). Thus, the majority of the proteome changes were indirect to the effects of CPEB1 in cell cycle, metabolism, etc. The authors further study the regulation by CPEB1 of just one of the candidates, MyoD1. Most of this characterization is similar to the analysis performed in the past for other CPEB1 targets in cell culture using reporters. The authors then determine CPEB1-phosphorylation upon activation of mSC and study the effect of MK5108 AurKa inhibitor and insulin on muscle regeneration.

This work has some interesting new observations, such as the proteome changes upon mSC activation or the differential expression of CPEB1. Other aspects are more expected, such as the regulation of MyoD1 mRNA by CPEB1, and the experimental approach has some limitations as to what could have been done in a more physiological context in SC activation. However this work has two critical limitations that limit the potential conclusions of this study. First, the CPEB1-RIP lacks a control (CPEB1 KO/KD or at least IgG pull down). In the absence of this control is not possible to identify the CPEB1 targets. This limitation contaminates most of the conclusions of this work. For the last part, the authors interpret the effects/phenotypes of MK5108 AurKa inhibitor and insulin as CPEB1 phosphorylation, however these treatments have many and very relevant effects independent of CPEB1, thus the conclusions are not correct. Other limitations are derived from the quality of the immunofluorescences and the limited characterizations of the Abs. Altogether, although certainly intriguing and, as an hypothesis very appealing, this work has fundamental technical limitations that make most of the conclusions of the work unsupported experimentally.

Critical points:

1- RIP-Seq (Fig 3) RIP has several major issues that could potentially invalidate the results. The main problem is that the targets are defined by enrichment of the IP vs input. It is well known that RIP, in the absence of parallel IgG or KD/KO controls, is not sufficient to determine the real targets of an RNA-BP (mostly but not only due to the unspecific binding of polysomes). The identity and expression levels of most of the identified target mRNAs point in the direction of non-specific immunoprecipitation. A CPEB1 western-blot control of the IPs would be required. Identification of the CPEs in the coimmunoprecipitated transcripts would be a proper quality control. Do the 39 CPEB1 regulated proteins in ASCs contain CPEs?

2- MK5108 AurKa inhibitor and insulin treatment have many other effects, in cell cycle, metabolism etc, in addition to CPEB1 phosphorylation. Thus it is incorrect to interpret the phenotypes of these treatments (Fig 7) as resulting from CPEB1 phosphorylation modulation.

Major points

1- Antibody specificity: Some of the most relevant new findings in this study are derived from

immunofluorescence analysis with CPEB1 and P-CPEB1 Abs. The characterization of the specificity of these Abs should be significantly improved (Even more so in light of the discrepancies between proteomics and immunofluorescence during mSC activation). For example, it would be more convincing to show that the CPEB1 band in western blot disappears in the CPEB1 KD (or a KO)(rather than relying in overexpression). For the phosphospecific, co-IP with CPEB1 Abs followed by western blot with P-CPEB1 Abs, with and without phosphatase treatments would be much more convincing that the western showed.

2- There are some KD/KO controls and quantitative analyses missing in the immunofluorescences. Subcellular (including granules/soluble) localization of CPEB1 would require much better resolution images and more accurate quantitative analyses.

3- siRNAs specificity should be addressed by overexpressing non-targetable CPEB1 mRNA to show specificity.

4- It is not clear why in some experiments the authors use fiSC instead of ASC, when CPEB1 is more expressed in the later and fiSC seem to be an artificial intermediate between ASCs and QSCs.

5- Figures 5-6: FISH of an mRNA not targeted by CPEB1 would be required as control. With the data/pictures shown it is not possible to claim co-localization.

6- What evidence do that authors have that the Ab is indeed binding and neutralizing CPEB1 inside the cells?

Minor Points

- Figure 1. Discrepancies between mRNA and protein levels could also be explained by changes in protein stability, as has been previously described for CPEB1. If the authors want to claim translational regulation further experiments would be required. Are global translation rates different in QSC and ASC (using OP-puro for instance)?

- Figure 2. Is CPEB1 localization nuclear?

- CPEB1 has been shown to have anti-proliferative effects and is regarded in several tumor types as a tumor suppressor; however, data here suggests that CPEB1 positively regulates cell proliferation. It will be worth to discuss this apparent controversy

Reviewer #4 (Remarks to the Author):

Comments:

CPEB1 directs muscle stem cell activation by reprogramming the translational landscape

Zeng et al, 2020

Summary

In this report, the authors use in situ fixation to preserve QSCs in combination with freshly isolated and cultured ASCs in combination with proteomic approaches to map changes in the proteomic landscape of SCs during quiescence, exit from quiescence and activation. They show discordance between transcriptomic and proteomic changes pointing towards translation as a major pathway being post-translationally activated upon SC activation. the authors subsequently focus on CPEB1, which controls translation of CPE containing transcripts. CPEB1 is upregulated in activated SCs and knock-down of CPEB1 reduces SC activation and proliferation. They further show that CPEB1 when phosphorylated by Aurka, improves the translation and protein expression of Myod1. This work reveals a new level of control of Myod1 content during SC activation, however, unraveling the role of CPEB1 in the SC phenotype should be better documented.

Results

Figure 1. It is unclear how the data were analyzed since only duplicates were provided and methods refer to the use of T-tests for statistical analysis of the data. Proper statistics and number of replicates are required. Please clarify this throughout the manuscript. Moreover, the lengthy description on what the data 'suggest' reads a bit as being redundant, since there is ample functional follow up work in the rest of the manuscript. Even though the use of cultured activated SCs is an elegant approach, the cells are cultured under high mitogenic conditions with ample nutrient availability. The latter conditions will significantly impact on the proteomic landscape of the SCs. Can the authors isolate SCs from damaged muscle to show similar (proteomic) changes between QSCs and in vivo ASCs?

Figure 2 (suppl. figure 3a). The description on what happens with CPEB1 under damaged/activated conditions and the interpretation of the siRNA experiments is not clear. First, quantifications of CPEB1 content (definitely on isolated SCs following damaged conditions in vivo) (Suppl 3a - fig 2b) are required. Moreover, since single fibers are isolated for the knock-down experiments, are the SCs not already activated before knockdown is achieved? The authors in fact refer to this issue at p12, line 17 'siRNA will require 16-24hours to function.' How can the authors conclude that loss of CPEB1 impairs 'activation' of SCs? In this respect, the authors would need to show other markers of SC behavior, next to increased SC cycling (edu), for instance Pax7/MyoD/Myf5, so the reader can understand whether differentiation/commitment/etc is affected or not. The authors in fact refer to this issue at p12, line 17 'siRNA will require 16-24hours to function.'

Figure 3. It is not clear why the authors use fiSCs for the RIP-seq experiment, since CPEB1 content is increasing only in ASCs. Can the authors explain the rationale for these experiments?

Figure 5. Do the authors have any way to show that the Ab was actually blocking CPEB1?

Figure 8: the authors should explain how they identified the 29 genes that are 'specifically' interacting with the CPEB1. What were the criteria? Why does the mutated have more interacting proteins than the wt? Unfortunately, this is not clear from the methods.

4. Figure 7. In order to show that insulin acts on pCPEB1 through Aurka and not via other pathways, the authors would need to combine insulin plus MK5108. While this is shown in figure 7, no data are included (suppl Fig 9 only shows single treatments). Also, the insulin experiment in vivo is confusing. Mice are injected with insulin twice, once right after injury and once one day after injury, but one also expects insulin levels to rise after eating. Do these injections lead to higher activation of insulin signaling? Can the authors explain in more detail the (technical) rationale behind this experiment based on previous papers/own observations? In line with this, how do the author reconcile their own work in relation to previous data that have suggested that insulin promotes SC behavior in an mTORC1 dependent manner? Finally, it's unclear how the authors provide evidence for increased self-renewal based on PAX7 staining only.

Minor comments:

1. Both in abstract as well in introduction and discussion, some sentences were not very well structured and some part were difficult to read/to follow. The introduction was not really boosting my interest and enthusiasm for the work-to-come, so the authors should consider rewriting.

2. Figure 2, fig. S3. How did the authors set the threshold for CPEB high and low? What is used for total SC total number? Pax7, MyoD? Can the authors explain why different markers are used? Can the authors provide a quantification of the knockdown efficiency. Because in fig.2g there seems to be CPEB still expressed.

3. Authors should consistently report n numbers of experiments. It would be good to show knock-down efficiency.

4. The discussion could benefit from a more in-depth discussion of the data when compared to previous work. It is currently merely limited to a repetition of the results section.

RESPONSE TO REVIEWERS' COMMENTS

General Responses

We thank the reviewers for their constructive comments regarding our manuscript, “CPEB1 directs muscle stem cell activation by reprogramming the translational landscape”. We have revised the text in accordance with the major and minor criticisms, and we are grateful for the suggestions that result in an improved manuscript.

In this response letter, we laid out our response to address the two major concerns of the reviewers: 1) the specificity of CPEB1 antibodies and 2) whether the effects of MK5108 or insulin are CPEB1-dependent. We found that the reviewers' remarks are largely addressable with the inclusion of a few new experiments. In the past several months, we have generated new data to further support our claims in the manuscript. We have also included all the control experiments such as IgG RIP-seq, siCPEB1 RT-qPCR, CPEB1 and phosphorylated CPEB1 antibody specificity testing experiments, and a dual luciferase assay on 293T cells after CPEB1 antibody transfection. Altogether, these control experiments confirm the specificity of the CPEB1 and pCPEB1 antibodies. To further explore the mechanism of how MK5108 or insulin regulates SC behavior via CPEB1, we treated SCs with insulin/MK5108 with CPEB1 knockdown or CPEB1 phosphorylation-mimic mutant (T171D, S177D) overexpression, respectively. CPEB1 knockdown or overexpression of CPEB1 phosphorylation-mimic mutant (T171D, S177D) partially rescues the effect of insulin treatment or MK5108 treatment respectively. Together, our new data suggest that insulin or MK5108 regulates SC behavior in part via the CPEB1 pathway. We have revised our conclusion and provided a detailed discussion on insulin/MK5108 phenotypes in the revised manuscript.

Here, we provide point-to-point clarifications regarding issues raised by the reviewers below. We have also highlighted all the changes in the revised manuscript.

We believe that our revised manuscript will be of interest to the broad readership of *Nature Communications*.

Point-by-Point Response to Reviewers' Comments (reviewers' comments are italicized):

Reviewer #1 (Remarks to the Author):

1. *“This manuscript from Dr. Cheung’s lab describes the proteome of quiescent MuSCs and at early states of activation. The data shows a discordance between transcriptome and proteome, thus evidence for post transcriptional regulation of MuSC activation. The authors focus on CPEB1-a molecule with no known function in MuSCs. While CPEB1 regulates a significant part of the proteome the authors focus on CPEB1 role in Myod1 regulation.*

This is a extremely novel study, the experiments are performed to highest standard and the data has uncovered a new layer of regulation for MuSC activation.

Major Strengths:

1) *Provides a comprehensive characterization of proteins expressed in MuSCs during quiescence to activation transition.*

2) *Comparative analysis with transcriptome.*

3) *Both are a significant addition to the field and will provide scientists opportunity for many hypothesis forming questions.*

4) *The authors have identified CPEB1-this has no known function in MuSC regulation to date.*

5) *The authors go on to identify the mechanism of CPEB1-using both unbiased approach and then focusing onto Myod1. The mechanistic aspect of this study is comprehensive and conclusive.*

6) *Use of a blocking antibody to disrupt intracellular protein activity is a novel approach.*

7) *The use of in vivo, ex vivo and in vitro approaches are essential to provide the in vivo and mechanistic studies.”*

We thank the Reviewer for his/her strongly positive comments and acknowledging our findings are interesting and novel.

2. *“While there are always more experiments, a reviewer can ask. There are only a couple of issues that should be addressed- based on the authors conclusions.*

After siRNA CPEB1 the authors report that MuSCs don’t activate based on a reduction in fraction of MuSCs that enter S-Phase (in Figure 2 and S3), I would like to know whether these cells are stalled in G0, G1 or dying through apoptosis.”

To determine whether these cells are dying through apoptosis, we stained Cleaved-Caspase 3 in SCs after knocking down CPEB1. SCs are negative for Cleaved-Caspase 3 staining, confirming that these cells are not dying (Suppl. Fig. 5b). To determine whether these cells are stalled in G0/G1 stage, we cultured the cells with EdU for 72 hours after CPEB1 knockdown, a time point by which the effect of siRNA is diminishing. After 72 hours, cells eventually take up EdU in the CPEB1 knockdown group (Suppl. Fig. 5c, d), suggesting that CPEB1 knockdown does indeed causes SCs to stall in G0/G1.

3. *“In the text, when discussing figure 7, please state in the text that insulin and MK5108 was used to perturb muscle regeneration in vivo-for ease of reader.”*

We have revised the text and added a statement that insulin and MK5108 were used to perturb muscle regeneration.

4. *“The authors state that CPEB1 is required for a proper MuSC proliferation environment. The use of environment could be misleading.”*

We have revised the statement to CPEB1 is required for MuSC proliferation.

Reviewer #2 (Remarks to the Author):

1. *“In the present article entitled “CPEB1 directs muscle stem cell activation by reprogramming the translational landscape” Zeng et al. investigate in vivo the translational and proteomic fingerprints of muscle stem cells using a novel whole body perfusion fixation technique able to obtain the unmodified quiescent signature avoiding common sample processing modifications. The manuscript is well-written, figures are nice and very illustrative, and the results are very interesting. Nonetheless, my main concern is the lack of proper discussion of results in discussion section, as this current version resembles a results section, specially the first parts. Thus, before publication in Nature Communications, I suggest doing some edits. Here, I detail my suggestions (or questions) point-by-point.”*

We thank the Reviewer for his/her positive comments. We agree with the Reviewer that the discussion section is short. We have revised the discussion according to the Reviewer’s comments.

Abstract

2. *“- If muscle stem cells (MuSCs) and satellite cells (SC) correspond to the same subset of cells, then better use only one of the terms. Maybe authors could say: “In skeletal muscle, the muscle stem cells, also known as satellite cells (SC), are actively regulated in the quiescent state.” And then only use one of the abbreviated terms. Now, both terms are used to refer to the same type of cells.”*

We have revised the text and used satellite cell (SC) exclusively in the revised manuscript.

3. *“- In the sentence: “Using a whole mouse perfusion fixation approach,...” I would add: “Using a whole mouse perfusion fixation approach to obtain true quiescent SCs,...” otherwise, it was not clear in the abstract the objective of the whole mouse perfusion fixation approach.”*

We have added ‘to obtain bona fide quiescent SC, ...’ to this sentence.

4. *“- In a related manner, in the abstract authors should give more importance to the technique used in this study. Please include a sentence explaining what is able to achieve the whole body perfusion fixation technique to investigate SC cells. Currently it is not clear until the reader finished the introduction section.”*

We have revised the abstract accordingly.

Results:

5. *“Line 124 – Delete/substitute “dramatic”. If authors want to add an adjective referring the observed*

changes maybe “significant” would be more adequate. I think dramatic is present in other parts of the text too.”

We have deleted the word ‘dramatic’ in the revised manuscript.

6. *“-The full name of the abbreviation PFA is missing.”*

We have added the full name of “PFA”, paraformaldehyde, in the revised manuscript.

Discussion:

7. *“As I already mentioned, authors need to discuss more the findings obtained in this study with the existent studies published by other authors in discussion section. I suggest editing discussion section expanding it. For example the whole first page of discussion looks like results. Additionally, in this section is also missing a discussion about the methodology used, and the existent methodology used to study the quiescence-to-activation transition. If this is the first time that is done a study of this biological phenomenon using the whole body perfusion fixation this has to be clear in discussion.”*

We thank the Reviewer for pointing this out. We have expanded the discussion in the revised manuscript. The whole body perfusion fixation approach to preserve quiescent SCs is recently published (PMID: 32502396 and PMID: 33377022). Thus, we did not provide a detailed discussion on this approach in this manuscript.

Figures:

8. *“Figure 1a. I suggest editing a bit the diagram of section 1a. It is not very clear. Specify clearly what population of cells was obtained by FACS.”*

We have revised the diagram of Fig. 1a and clarified the figure legend in the revised manuscript.

Methods:

9 *“- It is not clear in this section what is the procedure used by the authors to obtain every subset of cells used in this study. For example, in the materials and methods it is not mentioned the term fSCs. Are this cells obtained without perfusion? Please ensure that all procedures are properly described in detail. The reader must be able to repeat all the experiments performed in this study reading the materials and methods.”*

We have provided a clear description of the experimental details in the method section in the revised manuscript.

10. *“- How many animals were used in the study? What is the n for the different experiments? Please detail it clearly in the text, in every experiment.”*

We have clarified the experimental details in Figure legends in the revised manuscript.

11. *“- Information about anesthesia (ej. Dose) and strategy of sacrifice need to be specified in the first*

section of methods.”

We have clarified the experimental details in the revised manuscript.

12. “- In “*Satellite cell isolation and culture*”, I suggest to include a Briefly, ... and explain a little bit the procedure used to isolate SCs.”

We have added a brief FACS-isolation procedure in the method section in the revised manuscript.

13. “- As fixation by perfusion is the main technique used in this study I suggest to include an independent section in materials explaining the procedure used. Please locate it at the beginning of the section after the animals section to make the distribution linear for the reader.”

Our lab has recently published a detailed protocol of the perfusion approach in *Star Protocols* (Yue et al., 2020). We have provided a brief description in the method section in the revised manuscript.

14. “- Please include PEAKS software version.”

We have added the PEAKS software version in the revised manuscript.

15. “- In the section “*Mass spectrometry and label-free quantification of mass spectrometry data.*” More information about the parameters used in Tims TOF instrument are missing. Authors should provide enough details to allow readers to repeat the analysis.”

We have added a detailed description in the method section in the revised manuscript.

16. “- Similarly, no information about the database search for proteomics data in PEAKS was detailed. Please mention the database used, the tolerance error for the parent ion and for fragments, as well as the FDR.”

We have added a detailed description in the method section in the revised manuscript.

17. “- I suggest editing the title “*Mass Spectrometry and qRT-PCR on C2 cells after mVenus antibody immunoprecipitation.*” to something like “*Preparation of C2 cells after mVenus antibody immunoprecipitation.*”

We have revised this title in the revised manuscript.

Refs.

18. “I could not find ref 20 cited in the text.”

We thank the Reviewer for pointing this out. We have added reference 20 in the revised manuscript.

19. “- I would also suggest to include a conclusions section,”

We have added a brief conclusion in the first part of the discussion in the revised manuscript.

Reviewer #3 (Remarks to the Author):

1. “*The work by Zeng et al approaches the contribution of CPEB1 to translational regulation of muscle SC activation.*”

Following previous transcriptomic studies from the group (in QSCs, fiSCs and activated SCs), now the authors perform proteomic analyses in the same 3 cell types. In addition to the information pertaining the proteome changes during mSC activation, comparison of mRNA levels and proteome provides an indirect measure of potential translation regulation (and or Protein stability, see below). Following this observation, the authors focus in CPEB1 finding that this protein is overexpressed upon mSC activation (proliferation). Depletion of CPEB1 (siCPEB1) is associated with lower proliferation (cell division). Then the authors perform CPEB1-RIP in fiSC and proteomics on siCPEB1 SC. The authors identify over five hundred proteins affected in the CPEB1KD. Crossing this list with the CPEB1 RIP (With all the limitations noted below) the authors identify 39 CPEB1 regulated proteins in ASCs (of which 9 are related to translation). Thus, the majority of the proteome changes were indirect to the effects of CPEB1 in cell cycle, methabolism, etc. The authors further study the regulation by CPEB1 of just one of the candidates, Myod1. Most of this characterization is similar to the analysis performed in the past for other CPEB1 targets in cell culture using reporters. The authors then determine CPEB1-phosphorylation upon activation of mSC and study the effect of MK5108 AurKa inhibitor and insulin un muscle regeneration.

This work has some interesting new observations, such as the proteome changes upon mSC activation or the differential expression of CPEB1. Other aspects are more expected, such as the regulation of Myod1 mRNA by CPEB1, and the experimental approach has some limitations as to what could have been done in a more physiological context in SC activation.”

We thank the Reviewer for his/her positive comments that our proteome changes upon QSC activation or alternation of CPEB1 expression are interesting. However, we respectfully disagree with the Reviewer that the regulation of Myod1 mRNA by CPEB1 is expected. We acknowledge that CPEB1 regulation on its targeted transcript translation via CPEs has previously been demonstrated. However, the function of CPEB1 regulating Myod1 protein expression upon activation of quiescent SCs has never been reported. Importantly, published studies on Myod1 mRNA regulation are more focused on the inhibition of translation to maintain SC quiescence (Hausburg et al., 2015, PMID: 25815583; Morree et al., 2017, PMID: 29073096). Our study thus provides a new angle that SC activation is promoted by enhancing CPEB1-mediated Myod1 mRNA translation.

2. “*However this work has two critical limitations that limit the potential conclusions of this study. First, the CPEB1-RIP lacks a control (CPEB1 KO/KD or at least IgG pull down). In the absence of this control is not possible to identify the CPEB1 targets. This limitation contaminates most of the conclusions of this work.*”

In the revised manuscript, we have provided the western blot data (Suppl. Fig. 2a-d) showing that the CPEB1 antibody is specific. To address the Reviewer’s concern, we have provided an IgG pull-down control (Suppl. Fig. 6b and Suppl. Table 3) and showed that indeed our CPEB1-RIP

experiment is specific. We have also provided the CPEB1 IP western blot data and confirmed that the CPEB1 antibody can pull down CPEB1 protein specifically (Suppl. Fig. 6a).

3. “For the last part, the authors interpret the effects/phenotypes of MK5108 AurKa inhibitor and insulin as CPEB1 phosphorylation, however these treatments have many and very relevant effects independent of CPEB1, thus the conclusions are not correct.”

We agree with the Reviewer that MK5108 and insulin have many effects other than CPEB1 phosphorylation. We have revised the conclusion and suggested that the effects (Fig. 7 and Suppl. Fig 14) are partially via CPEB1 phosphorylation in the revised manuscript. To address the Reviewer’s concern, we showed that knocking down CPEB1 or overexpressing CPEB1 (T171D, S177D) could rescue the phenotype of insulin or MK5108 treatment respectively (Suppl. Fig 14a-d). We have also revised the discussion on these phenotypes to clarify that the phenotype of insulin/MK5108 is partially mediated through CPEB1.

4. “Other limitations are derived from the quality of the immunofluorescences and the limited characterizations of the Abs. Altogether, although certainly intriguing and, as an hypothesis very appealing, this work has fundamental technical limitations that make most of the conclusions of the work unsupported experimentally.”

We thank the Reviewer for his/her positive comments and acknowledging some of our findings are interesting and appealing. In the updated manuscript, we have included new panels for antibodies characterization using siRNA or λ protein phosphatase approach (Suppl. Fig 2c-f and Suppl. Fig. 10c) to show that both CPEB1 and pCPEB1 antibodies are specific. We have also shown the specificity of CPEB1 antibodies in Suppl. Fig. 2a, b and Suppl. Fig 10a, b using an overexpression approach. Together with the CPEB1-RIP and Myod1 CPE mutagenesis data, we have used multiple approaches to show that CPEB1 binds to the CPEs of Myod1 mRNA and regulates its translation. We believe the updated manuscript has addressed the Reviewer’s concerns regarding the technical limitations that he/she mentioned.

Critical points:

5. “RIP-Seq (Fig 3) RIP has several major issues that could potentially invalidate the results. The main problem is that the targets are defined by enrichment of the IP vs input. It is well known that RIP, in the absence of parallel IgG or KD/KO controls, is not sufficient to determine the real targets of an RNA-BP (mostly but not only due to the unspecific binding of polysomes). The identity and expression levels of most of the identified target mRNAs point in the direction of non-specific immunoprecipitation”

We apologize that we did not include the IgG control data in the initial submission. The IgG control pulled down a small portion of RNAs that associated with CPEB1 (Suppl. Fig. 6b). We have reanalyzed our data and excluded the few mRNAs that were pulled down by IgG controls in the revised manuscript (Fig. 3b-e). As expected, the exclusion of IgG bound RNAs does not affect our conclusion.

“A CPEB1 western-blot control of the IPs would be required.”

We have provided a CPEB1 western blot control for the IPs in Suppl. Fig. 6a and confirmed that the CPEB1 antibody is specific to pull down CPEB1 protein.

“Identification of the CPEs in the coimmunoprecipitated transcripts would be a proper quality control.”

We showed that 56.1% CPEB1’s targets contain CPEs while only 37.6% of IgG bound transcripts contain CPEs (Figure 3c).

“Do the 39 CPEB1 regulated proteins in ASCs contain CPEs?”

We thank the Reviewer for pointing this out. When we included the IgG control data and exclude the IgG bound transcripts, we only obtain 16 genes that are targets of CPEB1 and upregulated by CPEB1 during SC activation. We found that all these genes contain CPEs (Fig. 4h, i). This confirms that CPEB1 IP is specific and our analysis is reliable in identifying the CPEB1 targeted mRNAs in SCs.

6. *“- MK5108 AurKa inhibitor and insulin treatment have many other effects, in cell cycle, metabolism etc, in addition to CPEB1 phosphorylation. Thus it is incorrect to interpret the phenotypes of these treatments (Fig 7) as resulting from CPEB1 phosphorylation modulation.”*

We agree with the Reviewer that the MK5108 and insulin treatment have many other effects that are independent of CPEB1. As discussed in point #3, to address the Reviewer’s concern, we used a rescue experiment approach by knocking down CPEB1 with insulin treatment and overexpressed CPEB1 (T171D, S177D) with MK5108 treatment. Knocking down CPEB1 decreases insulin’s effect on SC activation whereas overexpressing CPEB1 (T171D, S177D) rescues MK5108-induced defects on SC activation (Suppl. Fig. 14a-d). These suggest that the phenotype we observed using insulin/MK5108 is at least in part mediated by CPEB1. We have revised the discussion of Figure 7 and clarified that the phenotype of insulin/MK5108 treatment is partially mediated by CPEB1.

Major points

7. *“1- Antibody specificity: Some of the most relevant new findings in this study are derived from immunofluorescence analysis with CPEB1 and P-CPEB1 Abs. The characterization of the specificity of these Abs should be significantly improved (Even more so in light of the discrepancies between proteomics and immunofluorescence during mSC activation). For example, it would be more convincing to show that the CPEB1 band in western blot disappears in the CPEB1 KD (or a KO)(rather than relying in overexpression). For the phosphospecific, co-IP with CPEB1 Abs followed by western blot with P-CPEB1 Abs, with and without phosphatase treatments would be much more convincing that the western showed.”*

We thank the Reviewer’s suggestion. We agree that the provide antibody specificity control experiment will strengthen our results. We have provided the immunostaining and western blot data (Suppl. Fig. 2a, b) showing that the CPEB1 antibody is indeed specific through overexpression of CPEB1 in 293T cells. Despite this, to address the Reviewer’s concern, we also

performed qRT-PCR, western blot, and immunostaining on primary myoblasts or SCs after CPEB1 siRNA transfection to confirm CPEB1 antibody specificity (Suppl. Fig. 2c-f). To validate the pCPEB1 antibody specificity, we have overexpressed wild-type or the phosphorylation site mutation CPEB1 into 293T cells. The pCPEB1 band was only shown in the wild-type CPEB1 overexpression group (Suppl. Fig. 10b). We also performed pCPEB1 western blot after phosphatase treatment to confirm the antibody specificity (Suppl. Fig. 10c). We believe our additional data further confirm that the antibodies are specific.

8. “2- There are some KD/KO controls and quantitative analyses missing in the immunofluorescences. Subcellular (including granules/soluble) localization of CPEB1 would require much better resolution images and more accurate quantitative analyses.”

We thank the Reviewer for pointing this out. We have provided more quantitative analysis on CPEB1 knockdown experiments in the revised manuscript (Suppl. Fig. 2c-f). We have provided the relative fluorescence units of CPEB1 to quantitatively measure the CPEB1 level (revised Fig. 2c). We have also provided magnified images and additional images of CPEB1 with DAPI to show the localization of CPEB1 (revised Fig. 2b). In Fig. 5c, d, and Fig. 6a, we have provided the magnified images to show the CPEB1 cytoplasmic colocalization with MyoD1 mRNA smFISH signals. We believe these images address the Reviewer’s concern about the resolution of the images.

9. “3- siRNAs specificity should be addressed by overexpressing non-targetable CPEB1 mRNA to show specificity.”

We thank the Reviewer for pointing this out. To address the Reviewer’s concern, we overexpressed the non-targetable CPEB1 (6 of the 19 targeted sequence is mutated) into 293T cells with siCPEB1 knockdown. As expected, CPEB1 siRNA can only target wild type CPEB1 to decrease the mRNA and protein level of CPEB1 but not the non-targetable CPEB1 (Suppl. Fig. 2g, h). These results confirm the specificity of siCPEB1.

10. “4- It is not clear why in some experiments the authors use fiSC instead of ASC, when CPEB1 is more expressed in the later and fiSC seem to be an artificial intermediate between ASCs and QSCs.”

We apologize that we did not clearly explain the rationale of using fiSC or ASC in some experiments and we have clarified the rationale of all the experiments in the revised manuscript. However, we respectfully disagree with the Reviewer that fiSC is an artificial intermediate state between ASCs and QSCs. The focus of this manuscript is the activation of quiescent SCs, which is an early time point. Multiple studies, including our previous work, have demonstrated that freshly isolated SCs (fiSCs) have acquired activation signatures (Machado et al., 2017, PMID: 29141227; Velthoven et al., 2017, PMID: 29141228; Yue et al., PMID: 32502396), representing the early activated SCs. Comparing with QSCs *in vivo*, fiSCs have not entered into the cell cycle but are early-activated SCs, while the activated SCs (named ASCs) in this study are fully activated, proliferating SCs 2 days post-isolation, cultured *in vitro*. FiSCs and ASCs are different from the perspective of the cell cycle state. Of note, previous studies mentioned above have suggested that fiSCs possess some levels of quiescent SCs signature, whereas ASCs do not. Since the focus of this manuscript is to understand the effect of CPEB1 during the activation process of quiescent SCs and we are trying to dissect the downstream effector of the CPEB1-mediated SC activation, we believe the early-activated SCs is an appropriate cell state for comparison and analysis.

11. “5- Figures 5-6: FISH of an mRNA not targeted by CPEB1 would be required as control.”

We thank the Reviewer for pointing this out. The specificity of the FISH probes has been tested and published in our previous study (Yue et al., 2020, PMID: 32502396). To address the Reviewer’s concern, we have provided a control experiment of *GAPDH* smFISH (Suppl. Fig. 8a, b).

“With the data/pictures shown it is not possible to claim co-localization.”

We respectfully disagree with the Reviewer’s comment that the data is not possible to claim co-localization. Besides *Myod1* mRNA smFISH (Fig. 5d), we have used multiple approaches including CPEB1 RIP-seq (Fig. 5a), CPEB1 RIP qRT-PCR (Fig. 5b), and CPE mutagenesis (Fig. 5f) to show that CPEB1 directly binds and regulates *Myod1* mRNA translation via CPEs. Nonetheless, to address the Reviewer’s concern, we have replaced the statement of ‘co-localization’ to ‘association’ in the revised manuscript.

12. “6- What evidence do that authors have that the Ab is indeed binding and neutralizing CPEB1 inside the cells?”

We thank the Reviewer for pointing this out. As discussed in point #4, we have multiple data (Suppl. Fig. 2) to show that the CPEB1 antibody is specific. Nonetheless, to address the Reviewer’s concern, we performed a dual luciferase assay on 293T cells after transfection with corresponding vector (flag-CPEB1, *Myod1*-mRNA 3’UTR contained luciferase vector and luciferase control vector) and CPEB1 antibody. Transfection of the CPEB1 antibody neutralized the increased luciferase signal by CPEB1 overexpression (Suppl. Fig. 8c). This confirms that the CPEB1 antibody could indeed neutralize CPEB1 protein and disrupt its function.

Minor Points

13. “- Figure 1. Discrepancies between mRNA and protein levels could also be explained by changes in protein stability, as has been previously described for CPEB1. If the authors want to claim translational regulation further experiments would be required. Are global translation rates different in QSC and ASC (using OP-puro for instance)?”

We agree with the Reviewer that the discrepancies between mRNA and protein levels could be caused by changes of protein stability by CPEB1 and we cannot exclude the function of CPEB1 on mRNA stability regulation, which might play an important role during SC activation. We acknowledge that this point is a limitation of the current study.

Regarding the difference between the global translation rates in QSCs and ASCs, it is a very interesting point. Our proteomic data comparison between QSCs, fiSCs, and cASCs showed some translation-related proteins are upregulated in fiSCs, suggesting translation is enhanced during SC activation (Suppl. Fig. 1d). It has previously been reported that general translation is suppressed in QSCs and it is relieved for rapid protein generation in ASCs (Zismanov et al., 2016, PMID: 26549106). Thus, we believe the global translation rate is different between QSCs and ASCs. However, the detailed information and the mechanisms behind it are beyond the scope of the current study.

14. “- *Figure 2. Is CPEB1 localization nuclear?*”

We thank the Reviewer for pointing this out. There are some CPEB1 nuclear signals in ASCs (Fig. 2b). Nuclear CPEB1 was reported to regulate alternative 3’UTR formation of its target (Bava et al., 2013, PMID: 23434754). In this manuscript, we focused on the mechanism of CPEB1 during the transition from quiescence to activation of QSCs. During this time point, CPEB1 is mainly located in the cytoplasm (Fig. 2b and 5c). The function of nuclear CPEB1 in fully activated SCs is beyond the scope of this study and we have included a discussion regarding the function of nuclear CPEB1 in the revised manuscript.

15. “- *CPEB1 has been shown to have anti-proliferative effects and is regarded in several tumor types as a tumor suppressor; however, data here suggests that CPEB1 positively regulates cell proliferation. It will be worth to discuss this apparent controversy*”

We thank the Reviewer for pointing this out. The cellular function of CPEB1 mainly depends on the targeted mRNAs and is cell type specific. For example, CPEB1 regulates oocyte maturation by targeting some of maternal mRNA in the oocyte and regulates proliferation by targeting cell cycle related genes (Racki et al., 2006, PMID: 17050619; Tay et al., 2003, PMID: 12815066; Kochanek et al., 2013, PMID: 23360795). As suggested by the Reviewer, we have provided a discussion about known CPEB1 functions in the revised manuscript.

Reviewer #4 (Remarks to the Author):

1. “*Comments:*”

CPEB1 directs muscle stem cell activation by reprogramming the translational landscape

Zeng et al, 2020

Summary

In this report, the authors use in situ fixation to preserve QSCs in combination with freshly isolated and cultured ASCs in combination with proteomic approaches to map changes in the proteomic landscape of SCs during quiescence, exit from quiescence and activation. They show discordance between transcriptomic and proteomic changes pointing towards translation as a major pathway being post-translationally activated upon SC activation. the authors subsequently focus on CPEB1, which controls translation of CPE containing transcripts. CPEB1 is upregulated in activated SCs and knock down of CPEB1 reduces SC activation and proliferation. They further show that CPEB1 when phosphorylated by Aurka, improves the translation and protein expression of Myod1. This work reveals a new level of control of Myod1 content during SC activation, however, unraveling the role of CPEB1 in the SC phenotype should be better documented.”

We thank the Reviewer for his/her positive comments on this manuscript. To address the Reviewer’s concern regarding the role of CPEB1 in the SC phenotype, we will address this comment as follows.

Results

2. *“It is unclear how the data were analyzed since only duplicates were provided and methods refer to the use of T-tests for statistical analysis of the data. Proper statistics and number of replicates are required. Please clarify this throughout the manuscript.”*

We thank the Reviewer for pointing this out. We have provided a detailed description in terms of statistical analysis of the data in the method section in the revised manuscript. We have also clarified the number of replicates for all the experiments in the revised manuscript.

“Moreover, the lengthy description on what the data 'suggest' reads a bit as being redundant, since there is ample functional follow up work in the rest of the manuscript.”

We thank the Reviewer’s comment and we have tried our best to make the descriptive parts more concise in the revised manuscript.

“Even though the use of cultured activated SCs is an elegant approach, the cells are cultured under high mitogenic conditions with ample nutrient availability. The latter conditions will significantly impact on the proteomic landscape of the SCs. Can the authors isolate SCs from damaged muscle to show similar (proteomic) changes between QSCs and in vivo ASCs?”

We thank the Reviewer for his/her positive comments. Despite the fact that cultured activated SCs are not exactly the same as ASCs isolated from injured muscle, cultured activated SCs have been widely used to study the activation signature or regulation in this field (Ryall et al., 2015, PMID: 25600643; Kimmel et al., 2020, PMID: 32198156). To address the Reviewer’s concern, we have isolated ASCs from injured muscle (iASCs) and compared the QSCs, fiSCs, cASCs and iASCs proteomes (Fig. 1b - e). We observed that the proteomic landscape of iASCs is very similar to cASCs (Fig. 1b - e) and the proteomic changes between QSCs and iASCs is similar to the comparison between QSCs and cASCs.

3. *“Figure 2 (suppl. figure 3a). The description on what happens with CPEB1 under damaged/activated conditions and the interpretation of the siRNA experiments is not clear. First, quantifications of CPEB1 content (definitely on isolated SCs following damaged conditions in vivo) (Suppl 3a - fig 2b) are required. Moreover, since single fibers are isolated for the knock down experiments, are the SCs not already activated before knock down is achieved? The authors in fact refer to this issue at p12, line 17 siRNA will require 16-24hours to function.’ How can the authors conclude that loss of CPEB1 impairs 'activation' of SCs?”*

We apologize that we did not previously show the CPEB1 content in injured ASCs and the interpretation of the siRNA experiment is confused. To address the Reviewer’s concern, we have provided the absolute CPEB1 immunofluorescence intensity in QSCs and ASCs rather than CPEB1 high/low comparison in the revised manuscript (revised Fig. 2c, revised Suppl. Fig. 3b).

We agree with the Reviewer’s concern that SCs have been early-activated during the isolation process before the siRNA fully function. This is a technical caveat in the field. In general, cell cycle re-entry is a definitive hallmark of SC activation. SCs enter the first S phase at around 24~36 hours after isolation (Liu et al., 2018, PMID: 30244867). Thus, we utilized the EdU incorporation assay to measure the number of SCs that have entered the S phase of the cell cycle as a conservative measurement of SC activation. We thus performed the siRNA transfection experiment right after the SC isolation and harvested the SCs or single fibers 24 or 36 hours after

siRNA transfection. The result suggests that upon CPEB1 knockdown, they have a delayed cell cycle entry by virtue of a reduced number of EdU positive SCs (Fig. 2d-h and Suppl. Fig. 3). To supplement this finding and confirm CPEB1 indeed regulate SC activation, we also used an antibody transfection approach to show that CPEB1 regulates MyoD1 protein expression (Fig. 5g-j). Based on these data, we believe that CPEB1 is an important regulator of SC activation.

“In this respect, the authors would need to show other markers of SC behavior, next to increased SC cycling (edu), for instance Pax7/MyoD/Myf5, so the reader can understand whether differentiation/commitment/etc is affected or not. The authors in fact refer to this issue at p12, line 17 'siRNA will require 16-24hours to function.'”

We thank the Reviewer’s interest on the differentiation/commitment aspect of SCs. To address the reviewer interests, we quantified the percentage of EdU⁺MyoD⁺, EdU⁻MyoD⁺, EdU⁺MyoD⁻, and EdU⁻MyoD⁻ SCs and the data showed that CPEB1 functions to regulate SC activation but not SC cell fate commitment (Fig. 2h). To study whether CPEB1 loss results in premature differentiation, we stained for MyoG after CPEB1 knockdown. The data showed that CPEB1 knockdown does not alter SC differentiation (Suppl. Fig. 4f, g). Thus, in this manuscript, we summarized that CPEB1 regulates SC activation.

4. *“Figure 3. It is not clear why the authors use fiSCs for the RIP-seq experiment, since CPEB1 content is increasing only in ASCs. Can the authors explain the rationale for these experiments?”*

We apologize that we did not clearly explain the rationale for using fiSC for the RIP-seq. The focus of this manuscript is the early activation of quiescent SCs. Multiple studies have demonstrated that freshly isolated SCs (fiSCs) have acquired activation signatures (Machado et al., 2017, PMID: 29141227; Velthoven et al., 2017, PMID: 29141228), representing the early activated SCs. Comparing with QSCs *in vivo*, fiSCs are early-activated but they have yet to enter the cell cycle, while the activated SCs (named ASCs) in this study are already fully activated and proliferating (2 days after culture). FiSCs and ASCs are different from the perspective of their cell cycle states.

Since the focus of this study is to understand the effects of CPEB1 during early SC activation, we reason that it is better to use fiSCs to perform CPEB1 RIP-seq for the identification of CPEB1 targets. On the contrary, ASCs are fully activated, proliferating SCs. We believe this time point could be too late and could provide a different set of CPEB1 targets as compared to SC early activation. We have provided a detailed explanation of the rationale in the revised manuscript.

5. *“Figure 5. Do the authors have any way to show that the Ab was actually blocking CPEB1?”*

We thank the Reviewer for pointing this out. To address the Reviewer’s concern, we have performed a dual luciferase assay co-transfecting the CPEB1 plasmids as well as the CPEB1 antibody in 293T cells. Transfection of CPEB1 antibody abolished the increased luciferase signal caused by CPEB1 overexpression (Suppl. Fig. 8c). This confirms that the CPEB1 antibody does indeed neutralize CPEB1 protein and disrupts its function.

6. *“Figure 8: the authors should explain how they identified the 29 genes that are 'specifically' interacting with the CPEB1. What were the criteria? Why does the mutated have more interacting proteins than the wt? Unfortunately, this is not clear from the methods.”*

We have revised the method and laid out a detailed description on the strategy regarding the identification of interacting proteins of CPEB1 or phosphorylation sites mutated CPEB1. In brief, we overexpressed CPEB1-mVenus or CPEB1 (T171A, S177A)-mVenus in C2C12 cells followed by immunoprecipitation using mVenus antibody. By comparing to the IgG control, we identified the interacting proteins of CPEB1 or CPEB1 phosphorylation mutant (Fig. 6g-k). To identify the specific interacting partners of CPEB1 or CPEB1 phosphorylation mutant, we overlapped these protein groups and excluded the proteins that are bound by both CPEB1 and CPEB1 phosphorylation mutant (Fig. 6j). We identified 29 proteins that are specifically interacted with CPEB1 while 62 proteins that are specifically bound by the CPEB1 phosphorylation mutant. Interestingly, we observed that mutated CPEB1 have more interacting proteins. We are also puzzled by this finding and the mechanism behinds the interaction requires further investigation.

7. “4. Figure 7. In order to show that insulin acts on pCPEB1 through Aurka and not via other pathways, the authors would need to combine insulin plus MK5108. While this is shown in figure 7, no data are included (suppl Fig 9 only shows single treatments).”

We have provided new immunofluorescence data to show the levels of pCPEB1 with combined treatment (Suppl. Fig. 12i). The combined treatment result suggests that insulin indeed acts on pCPEB1 through the Aurka pathway.

“Also, the insulin experiment in vivo is confusing. Mice are injected with insulin twice, once right after injury and once one day after injury, but one also expects insulin levels to rise after eating. Do these injections lead to higher activation of insulin signaling?”

To target the SC in a more direct way and minimize the whole-body system effect, we injected insulin intramuscularly. We apologized that we did not provide a clear rationale for this experiment. To minimize the effect of endogenous insulin and the possible degradation of injected insulin, we injected insulin once more into the injured muscle one day after injury. By comparing between the insulin injected group and the control group, we believe that the SC behavior we observed depends on the insulin via intramuscular injections.

To address the Reviewer’s concern on whether the injection leads to a higher activation of insulin signaling, we sorted ASCs from injured muscle with different treatments and stained them for pCPEB1. The level of pCPEB1 is increased in insulin injected group (Suppl. Fig. 12j, k). Since insulin has many functions other than increasing CPEB1 phosphorylation, we also provided an additional rescue experiment where we treated the CPEB1 knockdown SCs with insulin (Suppl. Fig 14. c, d). This experiment confirmed that the function of insulin on SC activation is in part mediated by CPEB1 phosphorylation. We have revised our conclusion in Figure 7 and provided detailed discussions regarding insulin treatment.

“Can the authors explain in more detail the (technical) rationale behind this experiment based on previous papers/own observations? In line with this, how do the author reconcile their own work in relation to previous data that have suggested that insulin promotes SC behavior in an mTORC1 dependent manner?”

We have not published any papers on insulin-related treatments or mTORC1.

“Finally, it's unclear how the authors provide evidence for increased self-renewal based on PAX7

staining only.”

We agree with the reviewer that we cannot be 100% certain that SCs are self-renewed solely based on Pax7 staining. Nonetheless, the Pax7 positive cells are found on centrally nucleated fibers, an indication of regenerated fibers after muscle injury. We believe that the SCs associated with these centrally nucleated fibers are indeed self-renewed. To address the Reviewer’s concern, we revised our wording from “self-renewal” to “Pax7 positive SC number” in the revised manuscript.

Minor comments:

8. *“1. Both in abstract as well in introduction and discussion, some sentences were not very well structured and some part were difficult to read/to follow. The introduction was not really boosting my interest and enthusiasm for the work-to-come, so the authors should consider rewriting.”*

We thank the Reviewer for his/her suggestions. We have revised the corresponding sentences and re-structured the introduction and discussion in the revised manuscript.

9. *“2. Figure 2, fig. S3. How did the authors set the threshold for CPEB high and low?”*

We agree with the Reviewer that the high/low quantification is subjective. To address the Reviewer’s concern and provide objective and reliable quantification, we have provided the absolute immunofluorescence intensity of single cells (Fig. 2c and Suppl. Fig. 3b) in the revised manuscript.

“What is used for total SC total number? Pax7, MyoD? Can the authors explain why different markers are used?”

In the CPEB1 knockdown experiments (Fig. 2e, Suppl Fig. 3d – j), we used Syn4 (Syndecan 4) to label all the SCs since Syn4 is expressed both in QSCs and ASCs (Tanaka et al., PMID: 19265661). For the CPEB1 knockdown experiment on FACS-isolated SCs, we used nuclei staining DAPI to label the SCs since almost 100% of the cells isolated based on our FACS-SC isolation protocol (Liu et al., PMID: 26401916) are SCs (CD31⁻, CD45⁻, Sca1⁻, mVcam⁺).

“Can the authors provide a quantification of the knockdown efficiency. Because in fig.2g there seems to be CPEB still expressed.”

We have provided the siRNA efficiency data at the mRNA level and protein level (Suppl. Fig. 2c-f) in the revised manuscript.

10. *“3. Authors should consistently report n numbers of experiments. It would be good to show knockdown efficiency.”*

We thank the Reviewer for pointing this out. We have provided the replicate numbers in each of the Figure legends in the revised manuscript. We have also provided the siRNA knockdown efficiency was shown by RT-qPCR, western blot, and immunostaining (Suppl. Fig. 2c-f).

11. *“4. The discussion could benefit from a more in-depth discussion of the data when compared to*

previous work. It is currently merely limited to a repetition of the results section.”

We thank the Reviewer for pointing this out. We have provided a detailed discussion in the revised manuscript.

REVIEWER COMMENTS

Reviewer #2 (Remarks to the Author):

The authors have successfully addressed all my previous concerns, thus this reviewer recommends publication of this manuscript.

Reviewer #3 (Remarks to the Author):

The revised manuscript by Zeng et al clarifies some of the raised points. However the main limitations of the study remain unsolved.

1- The IgG control in sup. Fig6 highlights the critical relevance of performing this control in every single RIP experiment. Thus, enrichment over total mRNA is not a valid approach. As shown in suppl. Fig 6 the number of mRNAs enriched in the IgGs is equivalent to the CPEB1 pull down and the overlapping is around 30%. This is not a single experiment, to determine the specificity of the Ab as implied by the authors, but a necessary normalization-control in each RIP. All the analyses must be re-done normalizing by IgG.

2- The authors combine the MK5108 and insulin treatments with phosphomimetic-CPEB1 overexpression and CPEB-KD to address the relative contribution of CPEB1 in this pathway. Although the authors conclude that these signaling events are mediated by CPEB1, the data shows (suppl fig 14) that phosphomimetic-CPEB1 overexpression equally stimulates SC activation independently of the presence or absence of MK5108 and that insulin activates SCs in both presence and absence of CPEB1. These experiments show that insulin and AurKa effects in SC activation are independent of CPEB1.

3- CPEB1 Ab specificity has been demonstrated, if for the overexpressed not the endogenous (please include MW markers and uncropped gels). However, the data for P-CPEB1 specificity is not convincing.

4- Subcellular localization image quality is not improved. Thus, it does not sustain the claims in the manuscript.

5- The FISH/CPEB1 colocalization is unclear. Much better images and quantification would be required. As it is, the differences between Myod1 and GAPDH are not evident.

Reviewer #4 (Remarks to the Author):

I have attached a file with my replies to the single comments below.

We thank the authors for addressing our comments in the revised manuscript.

We have included our comments to the replies of the authors below.

1. “Comments:

CPEB1 directs muscle stem cell activation by reprogramming the translational landscape

Zeng et al, 2020

Summary

In this report, the authors use in situ fixation to preserve QSCs in combination with freshly isolated and cultured ASCs in combination with proteomic approaches to map changes in the proteomic landscape of SCs during quiescence, exit from quiescence and activation. They show discordance between transcriptomic and proteomic changes pointing towards translation as a major pathway being posttranslationally

activated upon SC activation. the authors subsequently focus on CPEB1, which controls translation of CPE containing transcripts. CPEB1 is upregulated in activated SCs and knock down of CPEB1 reduces SC activation and proliferation. They further show that CPEB1 when phosphorylated by Aurka, improves the translation and protein expression of Myod1. This work reveals a new level of control of Myod1 content during SC activation, however, unraveling the role of CPEB1 in the SC phenotype should be better documented.”

We thank the Reviewer for his/her positive comments on this manuscript. To address the Reviewer’s concern regarding the role of CPEB1 in the SC phenotype, we will address this comment as follows.

Results

2. “It is unclear how the data were analyzed since only duplicates were provided and methods refer to the use of T-tests for statistical analysis of the data. Proper statistics and number of replicates are required. Please clarify this throughout the manuscript.”

We thank the Reviewer for pointing this out. We have provided a detailed description in terms of statistical analysis of the data in the method section in the revised manuscript. We have also clarified the number of replicates for all the experiments in the revised manuscript.

T-tests are consistently used throughout the manuscript, while in many cases other statistical methods (ANOVA, adjusted p values) should be used. The authors also indicate in the Reporting summary that normality tests and adjustments for multiple comparisons are N/A for this manuscript. This is not true.

“Moreover, the lengthy description on what the data 'suggest' reads a bit as being redundant, since there is ample functional follow up work in the rest of the manuscript.”

We thank the Reviewer’s comment and we have tried our best to make the descriptive parts more concise in the revised manuscript.

This adaptation has not been done. Description of data in Figure 1 (p6-7) has not been adapted.

“Even though the use of cultured activated SCs is an elegant approach, the cells are cultured under high mitogenic conditions with ample nutrient availability. The latter conditions will significantly impact on the proteomic landscape of the SCs. Can the authors isolate SCs from damaged muscle to show similar (proteomic) changes between QSCs and in vivo ASCs?”

We thank the Reviewer for his/her positive comments. Despite the fact that cultured activated SCs are not exactly the same as ASCs isolated from injured muscle, cultured activated SCs have been widely used to study the activation signature or regulation in this field (Ryall et al., 2015, PMID: 25600643; Kimmel et al., 2020, PMID: 32198156). To address the Reviewer’s concern, we have isolated ASCs from injured muscle (iASCs) and compared the QSCs, fiSCs, cASCs and iASCs proteomes (Fig. 1b - e). We observed that the proteomic landscape of iASCs is very similar to cASCs (Fig. 1b - e) and the proteomic changes between QSCs and iASCs is similar to the comparison between QSCs and cASCs.

We thank the authors for adding this data.

3. “Figure 2 (suppl. figure 3a). The description on what happens with CPEB1 under damaged/activated conditions and the interpretation of the siRNA experiments is not clear. First, quantifications of CPEB1 content (definitely on isolated SCs following damaged conditions in vivo) (Suppl 3a - fig 2b) are required. Moreover, since single fibers are isolated for the knock down experiments, are the SCs not already activated before knock down is achieved? The authors in fact refer to this issue at p12, line 17 ‘siRNA will require 16-24hours to function.’ How can the authors conclude that loss of CPEB1 impairs ‘activation’ of SCs?”

We apologize that we did not previously show the CPEB1 content in injured ASCs and the interpretation of the siRNA experiment is confused. To address the Reviewer’s concern, we have provided the absolute CPEB1 immunofluorescence intensity in QSCs and ASCs rather than CPEB1 high/low comparison in the revised manuscript (revised Fig. 2c, revised Suppl. Fig. 3b).

We thank the authors for adding this data. Please clarify in the methods section how the IF quantification method was performed and which intensity parameter was used? Minor comment: Fig 2c there is a missing h after 72 (x-axis)

We agree with the Reviewer’s concern that SCs have been early-activated during the isolation process before the siRNA fully function. This is a technical caveat in the field. In general, cell cycle re-entry is a definitive hallmark of SC activation. SCs enter the first S phase at around 24~36 hours after isolation (Liu et al., 2018, PMID: 30244867). Thus, we utilized the EdU incorporation assay to measure the number of SCs that have entered the S phase of the cell cycle as a conservative measurement of SC activation. We thus performed the siRNA transfection

experiment right after the SC isolation and harvested the SCs or single fibers 24 or 36 hours after siRNA transfection. The result suggests that upon CPEB1 knockdown, they have a delayed cell cycle entry by virtue of a reduced number of EdU positive SCs (Fig. 2d-h and Suppl. Fig. 3). To supplement this finding and confirm CPEB1 indeed regulate SC activation, we also used an antibody transfection approach to show that CPEB1 regulates Myod1 protein expression (Fig. 5gj).

Based on these data, we believe that CPEB1 is an important regulator of SC activation.

The authors use both MyoD expression as well as proliferation as two indicators of 'activation'. These are however two completely different features, and it is important that the authors pay attention to not mix up activation versus proliferation, OR clearly describe a rationale why conclusions are made about activation, based on proliferation.

“In this respect, the authors would need to show other markers of SC behavior, next to increased SC cycling (edu), for instance Pax7/MyoD/Myf5, so the reader can understand whether differentiation/commitment/etc is affected or not. The authors in fact refer to this issue at p12, line 17 'siRNA will require 16-24hours to function.'”

We thank the Reviewer’s interest on the differentiation/commitment aspect of SCs. To address the reviewer interests, we quantified the percentage of EdU+MyoD+, EdU-MyoD+, EdU+MyoD-, and EdU-MyoD- SCs and the data showed that CPEB1 functions to regulate SC activation but not SC cell fate commitment (Fig. 2h). To study whether CPEB1 loss results in premature differentiation, we stained for MyoG after CPEB1 knockdown. The data showed that CPEB1 knockdown does not alter SC differentiation (Suppl. Fig. 4f, g). Thus, in this manuscript, we summarized that CPEB1 regulates SC activation.

We thank the authors for adding this data.

4. *“Figure 3. It is not clear why the authors use fiSCs for the RIP-seq experiment, since CPEB1 content is increasing only in ASCs. Can the authors explain the rationale for these experiments?”*

We apologize that we did not clearly explain the rationale for using fiSC for the RIP-seq. The focus of this manuscript is the early activation of quiescent SCs. Multiple studies have demonstrated that freshly isolated SCs (fiSCs) have acquired activation signatures (Machado et al., 2017, PMID: 29141227; Velthoven et al., 2017, PMID: 29141228), representing the early activated SCs. Comparing with QSCs *in vivo*, fiSCs are early-activated but they have yet to enter the cell cycle, while the activated SCs (named ASCs) in this study are already fully activated and proliferating (2 days after culture). FiSCs and ASCs are different from the perspective of their cell cycle states.

Since the focus of this study is to understand the effects of CPEB1 during early SC activation, we

reason that it is better to use fiSCs to perform CPEB1 RIP-seq for the identification of CPEB1 targets. On the contrary, ASCs are fully activated, proliferating SCs. We believe this time point could be too late and could provide a different set of CPEB1 targets as compared to SC early activation. We have provided a detailed explanation of the rationale in the revised manuscript.

We thank the authors for clarifying.

5. *“Figure 5. Do the authors have any way to show that the Ab was actually blocking CPEB1?”*

We thank the Reviewer for pointing this out. To address the Reviewer’s concern, we have performed a dual luciferase assay co-transfecting the CPEB1 plasmids as well as the CPEB1 antibody in 293T cells. Transfection of CPEB1 antibody abolished the increased luciferase signal caused by CPEB1 overexpression (Suppl. Fig. 8c). This confirms that the CPEB1 antibody does indeed neutralize CPEB1 protein and disrupts its function.

We thank the authors for adding this data.

6. *“Figure 8: the authors should explain how they identified the 29 genes that are 'specifically' interacting with the CPEB1. What were the criteria? Why does the mutated have more interacting proteins than the wt? Unfortunately, this is not clear from the methods.”*

We have revised the method and laid out a detailed description on the strategy regarding the identification of interacting proteins of CPEB1 or phosphorylation sites mutated CPEB1. In brief, we overexpressed CPEB1-mVenus or CPEB1 (T171A, S177A)-mVenus in C2C12 cells followed by immunoprecipitation using mVenus antibody. By comparing to the IgG control, we identified the interacting proteins of CPEB1 or CPEB1 phosphorylation mutant (Fig. 6g-k). To identify the specific interacting partners of CPEB1 or CPEB1 phosphorylation mutant, we overlapped these protein groups and excluded the proteins that are bound by both CPEB1 and CPEB1 phosphorylation mutant (Fig. 6j). We identified 29 proteins that are specifically interacted with CPEB1 while 62 proteins that are specifically bound by the CPEB1 phosphorylation mutant. Interestingly, we observed that mutated CPEB1 have more interacting proteins. We are also puzzled by this finding and the mechanism behinds the interaction requires further investigation.

We thank the authors for clarifying.

7. *“4. Figure 7. In order to show that insulin acts on pCPEB1 through Aurka and not via other pathways, the authors would need to combine insulin plus MK5108. While this is shown in figure 7, no data are included (suppl Fig 9 only shows single treatments).”*

We have provided new immunofluorescence data to show the levels of pCPEB1 with combined

treatment (Suppl. Fig. 12i). The combined treatment result suggests that insulin indeed acts on pCPEB1 through the Aurka pathway.

We thank the authors for adding those data to the revised manuscript.

“Also, the insulin experiment in vivo is confusing. Mice are injected with insulin twice, once right after injury and once one day after injury, but one also expects insulin levels to rise after eating. Do these injections lead to higher activation of insulin signaling?”

To target the SC in a more direct way and minimize the whole-body system effect, we injected insulin intramuscularly. We apologized that we did not provide a clear rationale for this experiment. To minimize the effect of endogenous insulin and the possible degradation of injected insulin, we injected insulin once more into the injured muscle one day after injury. By comparing between the insulin injected group and the control group, we believe that the SC behavior we observed depends on the insulin via intramuscular injections.

To address the Reviewer’s concern on whether the injection leads to a higher activation of insulin signaling, we sorted ASCs from injured muscle with different treatments and stained them for pCPEB1. The level of pCPEB1 is increased in insulin injected group (Suppl. Fig. 12j, k). Since insulin has many functions other than increasing CPEB1 phosphorylation, we also provided an additional rescue experiment where we treated the CPEB1 knockdown SCs with insulin (Suppl. Fig 14. c, d). This experiment confirmed that the function of insulin on SC activation is in part mediated by CPEB1 phosphorylation. We have revised our conclusion in Figure 7 and provided detailed discussions regarding insulin treatment.

We thank the authors for adding those data to the revised manuscript.

“Can the authors explain in more detail the (technical) rationale behind this experiment based on previous papers/own observations? In line with this, how do the author reconcile their own work in relation to previous data that have suggested that insulin promotes SC behavior in an mTORC1 dependent manner?”

We have not published any papers on insulin-related treatments or mTORC1.

The authors have overlooked here that there is considerable literature evidence about insulin induced mTORC1 activation in satellite cells.

“Finally, it's unclear how the authors provide evidence for increased self-renewal based on PAX7 staining only.”

We agree with the reviewer that we cannot be 100% certain that SCs are self-renewed solely based

on Pax7 staining. Nonetheless, the Pax7 positive cells are found on centrally nucleated fibers, an indication of regenerated fibers after muscle injury. We believe that the SCs associated with these centrally nucleated fibers are indeed self-renewed. To address the Reviewer's concern, we revised our wording from "self-renewal" to "Pax7 positive SC number" in the revised manuscript.

Please also make this adaptation on p16, line 20.

Minor comments:

8. "1. Both in abstract as well in introduction and discussion, some sentences were not very well structured and some part were difficult to read/to follow. The introduction was not really boosting my interest and enthusiasm for the work-to-come, so the authors should consider rewriting."

We thank the Reviewer for his/her suggestions. We have revised the corresponding sentences and re-structured the introduction and discussion in the revised manuscript.

We thank the authors for adapting this.

9. "2. Figure 2, fig. S3. How did the authors set the threshold for CPEB high and low?"

We agree with the Reviewer that the high/low quantification is subjective. To address the Reviewer's concern and provide objective and reliable quantification, we have provided the absolute immunofluorescence intensity of single cells (Fig. 2c and Suppl. Fig. 3b) in the revised manuscript.

We thank the authors for adapting this.

"What is used for total SC total number? Pax7, MyoD? Can the authors explain why different markers are used?"

In the CPEB1 knockdown experiments (Fig. 2e, Suppl Fig. 3d – j), we used Syn4 (Syndecan 4) to label all the SCs since Syn4 is expressed both in QSCs and ASCs (Tanaka et al., PMID: 19265661). For the CPEB1 knockdown experiment on FACS-isolated SCs, we used nuclei staining DAPI to label the SCs since almost 100% of the cells isolated based on our FACS-SC isolation protocol (Liu et al., PMID: 26401916) are SCs (CD31-,CD45-,ScaI-,mVcam+).

We thank the authors for clarifying this.

"Can the authors provide a quantification of the knockdown efficiency. Because in fig.2g there seems to be CPEB still expressed."

We have provided the siRNA efficiency data at the mRNA level and protein level (Suppl. Fig. 2cf) in the revised manuscript.

We thank the authors for adapting this.

10. “3. *Authors should consistently report n numbers of experiments. It would be good to show knockdown efficiency.*”

We thank the Reviewer for pointing this out. We have provided the replicate numbers in each of the Figure legends in the revised manuscript. We have also provided the siRNA knockdown efficiency was shown by RT-qPCR, western blot, and immunostaining (Suppl. Fig. 2c-f).

We thank the authors for adapting this. We refer to our comment about the statistics above.

11. “4. *The discussion could benefit from a more in-depth discussion of the data when compared to previous work. It is currently merely limited to a repetition of the results section.*”

We thank the Reviewer for pointing this out. We have provided a detailed discussion in the revised manuscript.

We thank the authors for adapting this.

RESPONSE TO REVIEWERS' COMMENTS

General Responses

We thank the reviewers for their constructive comments regarding our manuscript, “CPEB1 directs muscle stem cell activation by reprogramming the translational landscape”. We have revised the text in accordance with the criticisms, and we are grateful for the suggestions that result in an improved manuscript. In particular, we are grateful that three out of four reviewers are satisfied with our revised manuscript with very minor additional comments or concerns.

In this response letter, we laid out our response to address the three comments of the Reviewers: 1) the IgG controls for the CPEB1-RIP-seq experiments, 2) improvement of cellular colocalization images and 3), the statistical analysis. We found that the reviewers' remarks are largely addressable by re-analysis of the RIP-seq data by normalizing to the IgG control for each single CPEB1-RIP experiment, as well as re-analyzing the multiple comparison and editing parts of the relevant methods section. We show that the main conclusion inclusive of the new data is consistent with our previous conclusion.

In this revision, we have also included three separate experiments of uncropped blots of all the western blots shown in this manuscript.

Here, we provide point-by-point clarifications regarding the issues raised by the reviewers below. We have also highlighted all of the changes in the revised manuscript.

We believe that our revised manuscript will be of interest to the broad readership of *Nature Communications*.

Point-by-Point Response to Reviewers' Comments (reviewers' comments are italicized):

Reviewer #2 (Remarks to the Author):

“The authors have successfully addressed all my previous concerns, thus this reviewer recommends publication of this manuscript.”

We thank the Reviewer for his/her support on our manuscript.

Reviewer #3 (Remarks to the Author):

“The revised manuscript by Zeng et al clarifies some of the raised points. However the main limitations of the study remain unsolved.”

We thank the Reviewer for their positive comments on our manuscript. To fulfill the Reviewer's concerns, we provide a point-by-point response as follows:

“1- The IgG control in sup. Fig6 highlights the critical relevance of performing this control in every single RIP experiment. Thus, enrichment over total mRNA is not a valid approach. As shown in suppl. Fig 6 the number of mRNAs enriched in the IgGs is equivalent to the CPEB1 pull down and the overlapping is around 30%. This is not a single experiment, to determine the specificity of the Ab as implied by the authors, but a necessary normalization-control in each RIP. All the analyses must be re-done normalizing by IgG.”

We thank the Reviewer for his/her comments on the analysis of CPEB1-RIP-seq data. In the revised manuscript, we re-analyzed the data by normalizing to the IgG control of every single experiment. We have captured 1561 transcripts that are CPEB1-associated (Figures 3b, c). The number of CPEB1-associated genes is altered while the major conclusion of this study remains unchanged. Moreover, over 60% of CPEB1-bound genes contain CPEs in their 3'UTR while less than 10% of IgG-bound genes contains CPEs (Figure 3d). This confirms our CPEB1 RIP-seq is reliable and we appreciate the Reviewer's suggestions. We have revised Figure 3 and 4 based on the new data analysis.

“2- The authors combine the MK5108 and insulin treatments with phosphomimetic-CPEB1 overexpression and CPEB-KD to address the relative contribution of CPEB1 in this pathway. Although the authors conclude that these signaling events are mediated by CPEB1, the data shows (suppl fig 14) that phosphomimetic-CPEB1 overexpression equally stimulates SC activation independently of the presence or absence of MK5108 and that insulin activates SCs in both presence and absence of CPEB1. These experiments show that insulin and AurKa effects in SC activation are independent of CPEB1.”

We agree with the Reviewer that phosphomimetic-CPEB1 overexpression stimulates SC activation in both the presence or absence of MK5108 and that insulin activates SCs in both the presence and absence of siRNA for CPEB1. However, Suppl. Fig 14b shows that phosphomimetic-CPEB1 overexpression partially rescues MK5108-induced delay in SC activation compared to the MK5108 only treatment, and MK5108 reduces the improved SC activation mediated by overexpression. We would like to clarify that MK5108 decreases CPEB1 phosphorylation level but does not affect the levels of phosphomimetic-CPEB1 in virally infected SCs. Suppl. Fig 14d shows that CPEB1 knockdown reduces the insulin-induced improvements in SC activation and that insulin rescues CPEB1 knockdown-mediated SC activation defects relative to the controls. Furthermore, our results shows that insulin improves CPEB1 phosphorylation (Suppl. Figs. 12h and i). Since transfection of siRNA targeting CPEB1 is not 100% efficient (Suppl. Figs. 2d-h), we believe our data supports our claim that insulin or MK5108 regulates SC activation at least partially through CPEB1.

“3- CPEB1 Ab specificity has ben demonstrated, if for the overexpressed not the endogenous (please include MW markers and uncropped gels). However, the data for P-CPEB1 specificity is not convincing.”

We respectfully disagree with the Reviewer that CPEB1 antibody specificity for endogenous CPEB1 protein is not demonstrated. In Suppl. Figs 2g and h, we have provided CPEB1 siRNA specificity and efficiency by real-time PCR, and a western blot of CPEB1 with or without the targeted sequence mutation. Afterwards, we performed CPEB1 knockdown experiments on cultured SCs followed by western blot and immunostaining (Suppl. Figs 2d-f). The data shows that the CPEB1 levels decreased following siRNA transfection. We believe these data supports the specificity of the CPEB1 antibody to detect endogenous CPEB1 protein. We feel disappointed that our data on the p-CPEB1 antibody specificity still did not convince the Reviewer. To validate pCPEB1 antibody specificity, we overexpressed either wild-type or the phosphorylation site mutation CPEB1 in 293T cells. The pCPEB1 band was only present in the wild-type CPEB1 overexpression group (Suppl. Fig. 10b). We also performed a pCPEB1 western blot after phosphatase treatment to confirm antibody specificity (Suppl. Fig. 10c) in cultured SCs. In this revision, we have provided three sets of separate uncropped gels in the resource data. We hope that the addition of these data will satisfy the Reviewer’s concern.

“4- Subcellular localization image quality is not improved. Thus, it does not sustain the claims in the manuscript.”

We have attempted to obtain the high quality and high-resolution images to fulfil the Reviewer’s concerns using the equipment and reagents available to us at our institution. In this manuscript, we have stated that the majority of CPEB1 is located in the cytoplasm, and we believe our data supports this claim (Figures 2b, e and 5c).

“5- The FISH/CPEB1 colocalization is unclear. Much better images and quantification would be required. As it is, the differences between Myod1 and GAPDH are not evident.”

For the colocalization of CPEB1 and Myod1 mRNA smFISH, we also attempted to provide the

high quality and high-resolution images using the equipment and reagents available to us at our institution to support our notion that CPEB1 targets Myod1 mRNA in the cytoplasm (Figures 5d and 6a). We also provided GAPDH smFISH as a control to show Myod1 mRNA does indeed significantly colocalize with CPEB1 protein (Suppl. Fig 8b). We observed some of the GAPDH smFISH signal overlapping with CPEB1 protein. We cannot exclude the possibility that co-staining methods could induce some false positives, however, the difference between GAPDH and Myod1 is significant. To support the imaging data, we have also provided the CPEB1 RIP-seq, CPEB1 RIP-real-time PCR and the dual luciferase data (Figures 5a, b and f respectively). The immunostaining data aims to provide a reference to demonstrate where in the cell CPEB1 targets Myod1 mRNA. We believe these observations supports our claim that CPEB1 targets Myod1 mRNA. We are willing to remove this data and believe its exclusion will not affect our main conclusion if the Reviewer still considers that the immunostaining data does not support our statements in this manuscript.

Reviewer #4 (Remarks to the Author):

“1. T-tests are consistently used throughout the manuscript, while in many cases other statistical methods (ANOVA, adjusted p values) should be used. The authors also indicate in the Reporting summary that normality tests and adjustments for multiple comparisons are N/A for this manuscript. This is not true.”

We thank the Reviewer for his/her comment on our manuscript. We have amended the related figure legends and methods in the revised manuscript. We have also updated the Reporting summary. We understand that ANOVA should be applied for multiple comparisons, while a pair-wise T-test is also suitable for comparison of different groups of treatments. In this manuscript, a pair-wise T-test was used to compare different groups of samples. To address the Reviewer’s concern, we also performed ANOVA analysis on data containing multiple samples (e.g., Suppl. Fig. 14b, d) and provided the analysis results of ANOVA and multiple comparisons in the Source data file.

“2. This adaptation has not been done. Description of data in Figure 1 (p6-7) has not been adapted.”

We thank the Reviewer for pointing this out. We have further trimmed the descriptive text of Figure 1 in the revised manuscript.

“3. We thank the authors for adding this data. Please clarify in the methods section how the IF quantification method was performed and which intensity parameter was used? Minor comment: Fig 2c there is a missing h after 72 (x-axis)””

We apologize that we did not provide this information. We quantified the IF images using the ZEN 2.5 lite software. We used the “mean” as the intensity parameter. We have clarified the information in the methods section. We thank the Reviewer for their reminder that there is a

missing “h” in Figure 2c. We have corrected and clarified these points in the revised manuscript.

“4. The authors use both MyoD expression as well as proliferation as two indicators of 'activation'. These are however two completely different features, and it is important that the authors pay attention to not mix up activation versus proliferation, OR clearly describe a rationale why conclusions are made about activation, based on proliferation.”

We apologize that we did not clearly explain the rationale for using Myod1 protein and EdU incorporation as activation indicators. The assumption is that all SCs are in quiescence at the start of the experiment. We use Myod1 protein expression as an indicator to distinguish early activated SCs (i.e., activated, but not proliferating). After SCs exit quiescence, SCs will re-enter the cell cycle (indicated by EdU incorporation, or cells in S phase) in the time window of around 36 hours. Should the activation process be impaired, SCs will take longer to enter the first cell cycle (i.e., fewer cells incorporating EdU). In the revised manuscript, we have paid attention to ensure the use of terms (i.e., activation or proliferation) is precise.

“5. The authors have overlooked here that there is considerable literature evidence about insulin induced mTORC1 activation in satellite cells.”

We thank the Reviewer for pointing this out. To address the Reviewer’s concern, we have revised the discussion part and referred to the related literature on mTORC1 activation in satellite cells.

“6. Please also make this adaptation on p16, line 20.”

We have amended the revised manuscript with this adaptation by changing “self-renewal SC” to “Pax7⁺ SC”.

REVIEWER COMMENTS

Reviewer #3 (Remarks to the Author):

1- A necessary normalization-control in each RIP. All the analyses must be re- done normalizing by IgG.

From the response of the authors it is unclear if they had originally performed the parallel IgG RIP in each of the experiment (but not used this data for normalization) and now they have; Or if they have normalized each original RIP to a single IgG control performed independently at a later time. If is the first case, then it is OK. If the authors have done the second, as can be inferred from their response, then it is not acceptable. IgG control must be run in parallel in each RIP experiment and for every condition.

2- phosphomimetic-CPEB1 overexpression equally stimulates SC activation independently of the presence or absence of MK5108 and that insulin activates SCs in both presence or absence of CPEB1.

The explanations provided do not solve the key point, whether Insulin and Aurka (Which are known to phosphorylate CPEB among many other proteins) functions are mediated by CPEB phosphorylation. The data, if anything, indicate that insulin and Aurka effects in SC activation are independent of CPEB1.

Thus, phosphomimetic-CPEB1 overexpression equally stimulates SC activation independently of the presence or absence of MK5108 and that insulin activates SCs in both presence or absence of CPEB1.

"3- CPEB1 Ab specificity has ben demonstrated, if for the overexpressed not the endogenous (please include MW markers and uncropped gels). However, the data for P-CPEB1 specificity is not convincing."

No new data or information on the specificity are provided, so my comment remains the same

"4- Subcellular localization image quality is not improved. Thus, it does tot sustain the claims in the manuscript."

5-" The FISH/CPEB1 colocalization is unclear. Much better images and quantification would be requires. As it is the differences between Myod1 and GAPDH are not evident."

The images and quantifications shown demonstrate that the resolution of the images are not enough to support the claims of the manuscript.

Reviewer #4 (Remarks to the Author):

The authors have addressed our comments. We congratulate them for their nice study.

RESPONSE TO REVIEWERS' COMMENTS

General Responses

We thank the reviewers for their constructive comments regarding our manuscript, “CPEB1 directs muscle stem cell activation by reprogramming the translational landscape”. We have revised the text in accordance with the criticisms, and we are grateful for the suggestions that result in an improved manuscript. We are grateful that three out of four reviewers are highly supportive for our manuscript to be published in *Nature Communications* and also appreciate the remaining comments raised by Reviewer #3, to improve our manuscript. In this response letter, we have laid out our response to address the comments of Reviewer #3 by 1) improvement of CPEB1 subcellular localization images and the exclusion of colocalization images of CPEB1 protein and Myod1 mRNA; 2) explanation of how MK5108 and insulin regulates SC activation at least in part through CPEB1 and 3), new ELISA data to support pCPEB1 antibody specificity. We found that the Reviewers' remarks are largely addressable by these explanations, the additional immunostaining, and the ELISA data.

Here, we provide point-by-point clarifications regarding the issues raised by Reviewer #3 below. We have also highlighted all the changes in the revised manuscript.

We believe that our revised manuscript will be of interest to the broad readership of *Nature Communications*.

Point-by-Point Response to Reviewers' Comments (reviewers' comments are italicized):

Reviewer #3 (Remarks to the Author):

“1- A necessary normalization-control in each RIP. All the analyses must be re- done normalizing by IgG. From the response of the authors it is unclear if they had originally performed the parallel IgG RIP in each of the experiment (but not used this data for normalization) and now they have; Or if they have normalized each original RIP to a single IgG control performed independently at a later time. If is the first case, then it is OK. If the authors have done the second, as can be inferred from their response, then it is not acceptable. IgG control must be run in parallel in each RIP experiment and for every condition..”

We thank the Reviewer for his/her comments on the analysis of CPEB1-RIP-seq data. We confirm that the CPEB1 RIP-seq experiments were performed in parallel with the IgG immunoprecipitation experiments.

“2- phosphomimetic-CPEB1 overexpression equally stimulates SC activation independently of the presence or absence of MK5108 and that insulin activates SCs in both presence or absence of CPEB1.

The explanations provided do not solve the key point, whether Insulin and Aurka (Which are known to phosphorylate CPEB among many other proteins) functions are mediated by CPEB phosphorylation. The data, if anything, indicate that insulin and AurKa effects in SC activation are independent of CPEB1.

Thus, phosphomimetic-CPEB1 overexpression equally stimulates SC activation independently of the presence or absence of MK5108 and that insulin activates SCs in both presence or absence of CPEB1.”

We agree with the Reviewer #3 that overexpression of phosphomimetic-CPEB1 stimulates SC activation in both the presence or absence of MK5108 and that insulin activates SCs in both the presence and absence of CPEB1 siRNA. However, we believe that insulin or MK5108 regulates SC activation at least in part through CPEB1 based on the following observations:

- 1) CPEB1 phosphorylation regulates SC activation (Suppl. Fig. 13h, i),
- 2) MK5108 or insulin treatment inhibits or promotes CPEB1 phosphorylation respectively (Suppl. Fig. 12i, k),
- 3) MK5108 or insulin treatment inhibits or promotes SC activation respectively (Fig. 7a, b),
- 4) Phosphomimetic-CPEB1 mimics CPEB1 phosphorylation and function due to the negative charge of the T171D, S177D mutation but not the actual phosphor group at the CPEB1 phosphorylation site (T171, S177). We understand that MK5108 can prevent the addition of the phosphor group to the phosphorylation site but it cannot remove the negative charge of the T171D, S177D mutation. Thus, MK5108 treatment decreased the endogenous levels of pCPEB1 but failed to inhibit phosphomimetic-CPEB1 function. Therefore, Suppl. Fig. 14b showed that phosphomimetic-CPEB1 overexpression partially rescued the MK5108-induced delay in SC activation compared to MK5108 only treatment, and that MK5108 dampened the effects of phosphomimetic-CPEB1 overexpression on SC activation,
- 5) Insulin improves CPEB1 phosphorylation, leading to increased percentage of pCPEB1,
- 6) siRNA-mediated knockdown did not remove all CPEB1 protein from SCs, showing that the knockdown efficiency was not 100%,
- 7) Insulin treatment after siCPEB1 transfection increases the percentage of pCPEB1. Hence, Suppl. Fig. 14d shows the CPEB1 knockdown reduces the insulin-induced improvements in SC activation and that insulin rescues the CPEB1 knockdown-mediated defects in SC activation relative to the controls.

Based on these observations, we conclude that insulin or MK5108 regulates SC activation at least in part through CPEB1. We believe our explanation can fulfil the Reviewer #3's concern.

"3- CPEB1 Ab specificity has ben demonstrated, if for the overexpressed not the endogenous (please include MW markers and uncropped gels). However, the data for P-CPEB1 specificity is not convincing.

No new data or information on the specificity are provided, so my comment remains the same"

We respectfully disagree with the Reviewer #3 that CPEB1 antibody specificity for endogenous CPEB1 protein was not demonstrated based on the following:

- 1) In Suppl. Fig. 2g and h, we have provided evidence of CPEB1 siRNA specificity and efficiency by real-time PCR, and a western blot of CPEB1 with or without the targeted sequence mutation and,
- 2) siRNA-mediated CPEB1 knockdown in cultured SCs led to decreased CPEB1 levels as shown by western blot and immunostaining (Suppl. Fig. 2d-f).

The MW markers and uncropped gels were shown in source data. Since we cut the membrane to blot different proteins (CPEB1 and GAPDH), the gel was shown by merging these images together. We believe these data strongly supports the specificity of the CPEB1 antibody to detect endogenous CPEB1 protein.

We feel disappointed that our data regarding pCPEB1 antibody specificity did not alleviate the Reviewer #3's concern. To validate pCPEB1 antibody specificity, we have performed loss and gain of function analysis and the following observations strengthen our claim that the pCPEB1

antibody specifically recognizes and binds to pCPEB1 protein:

- 1) To validate pCPEB1 antibody specificity, we overexpressed either flag tagged wild-type (wt) or the CPEB1 isoform carrying a mutation at the phosphorylation site (T171A, S177A) in 293T cells. The residue Alanine (A) cannot be phosphorylated. We only detected pCPEB1 band in the wild-type CPEB1 overexpression group (Suppl. Fig. 9b), supporting that the pCPEB1 antibody detects the phosphor group. Furthermore, the pCPEB1 protein band overlapped with the flag tag band, suggesting that this band is indeed CPEB1 protein. We also performed pCPEB1 immunostaining and showed that the pCPEB1 signals were only detected in 293T cells expressing flag-wtCPEB1. These signals also partially overlapped with flag tag signals (Suppl. Fig. 9a), which is expected as CPEB1 may not be phosphorylated in all cells. Based on these observations, we conclude that the pCPEB1 antibody can specifically recognize phosphorylated CPEB1.
- 2) The pCPEB1 western blot of (Suppl. Fig. 9c) cultured SCs treated with phosphatase provides further evidence of antibody specificity. Lambda phosphatase treatment decreased levels of phosphorylated CPEB1 but treatment could not completely remove pCPEB1 which was also reported in various studies (PMID: 10205174; PMID: 21051949). In this revision, we have provided three sets of separate uncropped gels in the resource data. Since we cut the membranes to blot with different antibodies (anti-pCPEB1 and GAPDH), the gel shown was formed by merging these images together. We believe that the addition of these data will satisfy the Reviewer #3's concern.
- 3) Finally, we also provide ELISA data (Suppl. Fig. 9d) to show the specificity and efficiency of the pCPEB1 antibody. The data demonstrated that the pCPEB1 antibody recognizes phosphor-peptides but not the non-phosphor-peptides even when the antibody was tested at different dilutions.

Based on these data, we firmly believe that the pCPEB1 antibody robustly recognizes phosphorylated CPEB1.

“4- Subcellular localization image quality is not improved. Thus, it does not sustain the claims in the manuscript.”

In this revision, we provide the high-resolution confocal images of QSCs, fiSCs and 4 hours cultured SCs demonstrating that CPEB1 and pCPEB1 are predominantly localized in the cytoplasm of QSCs and fiSCs whilst in 4 hours cultured SCs, some of the CPEB1 and pCPEB1 signals are detected in nuclei (Fig. 5d and 6a). This observation supports our claim that CPEB1 is located in the cytoplasm when the SCs are transitioning from quiescence to activation and is located in both the cytoplasm and nuclei of activated SCs.

“5- “ The FISH/CPEB1 colocalization is unclear. Much better images and quantification would be required. As it is the differences between Myod1 and GAPDH are not evident.”

The images and quantifications shown demonstrate that the resolution of the images are not enough to support the claims of the manuscript.”

We previously provided the CPEB1 protein immunostaining and Myod1 transcripts smFISH data to show the colocalization of CPEB1 and Myod1 mRNA in the cytoplasm. This smFISH data aimed to demonstrate where in the cell CPEB1 targets Myod1 mRNA. The Reviewer #3's concern was with regards to the resolution and the quantification method, so we have elected to remove

these smFISH data. We believe the exclusion of the smFISH data does not affect our main conclusion that CPEB1 drives SC activation by regulating Myod1 protein expression as we have previously provided the CPEB1 RIP-seq and CPEB1 RIP-real-time PCR data demonstrating CPEB1 can bind to Myod1 mRNA (Fig. 5a-c). Moreover, we also demonstrated CPEB1 regulates Myod1 protein expression using the dual luciferase assay and Myod1 protein immunostaining after CPEB1 antibody transfection (Fig. 5g-j). These data support the notion that CPEB1 regulates Myod1 protein expression to promote SC activation.

Reviewer #4 (Remarks to the Author):

“The authors have addressed our comments. We congratulate them for their nice study.”

We are very grateful for his/her support on our study.

REVIEWERS' COMMENTS

Reviewer #3 (Remarks to the Author):

Point 4 has been satisfactorily addressed.

RESPONSE TO REVIEWERS' COMMENTS

General Responses

We thank the reviewers for their constructive comments regarding our manuscript, "CPEB1 directs muscle stem cell activation by reprogramming the translational landscape". We have revised the text in accordance with the criticisms, and we are grateful for the suggestions that result in an improved manuscript. We are grateful that the editor helps to summarize the remaining concerns for our manuscript. In this response letter, we have laid out our response to address the remaining comments by 1) providing a comparison of CPEB1 or pCPEB1 expression during SC activation by western blot; 2) toning down the claims regarding the effects of insulin and Aurka on SC function through CPEB1 phosphorylation; 3) adjusting the conclusion regarding the manipulation of CPEB1 phosphorylation *in vivo* for regulating SC function, and 4) revising the text and figures following author checklist provided by the editor.

Here, we provide point-by-point clarifications regarding the remaining comments. We have also highlighted all the changes in the revised manuscript.

Point-by-Point Response to Reviewers' Comments (reviewers' comments are italicized):

"2 – They agreed with Reviewer #3 that effects of insulin are likely mediated by CPEB1-independent mechanisms. Supp Fig 14c-d show that insulin stimulated SC proliferation even more (or at least equal) when CPEB is knocked down. This can't be explained by KD efficiency. Current data for insulin/Aurkb-pCEBPI-SC activation is correlative, and it is possible that insulin/Aurka act through other targets."

We agree with the Reviewers' comments, and we toned down the claim that insulin/Aurka regulates SC function through CPEB1 phosphorylation and also discuss the other possibilities of how insulin/Aurka regulates SC function in the discussion section.

"3 – They are for the most part satisfied with assessment of anti-pCPEB antibody specificity, but not clear why changes were not assessed in endogenous pCPEB1 by Western blot in activated SCs."

We thank the Reviewers for their kind comments. In the final revision, we provided the western blot data of CPEB1 and pCPEB1 on freshly isolated SCs and activated SCs sorted from 3 days post injured muscles. The total protein of SCs is limited, therefore, to obtain satisfactory western blot data, we must isolate SCs from several mice for each biological replicate. Thus, in the earlier version of the manuscript, we opt to provide the immunostaining data to assess CPEB1 and pCPEB1 expression levels during SC activation.

"5 – They felt removal of FISH/CEBPI co-localization data is okay."

We thank the Reviewers for their kind comments.